# The HIF target MAFF promotes tumor invasion and metastasis through IL11 and STAT3 signaling

Eui Jung Moon[1,2], Stephano S. Mello[1,3], Caiyun G. Li[1], Jen-Tsan Chi [ID] [4], Kaushik Thakkar [ID] [1], Jacob G. Kirkland [ID] [5,6], Edward L. Lagory[1], Ik Jae Lee[7], Anh N. Diep[1], Yu Miao[1], Marjan Rafat [ID] [1,8], Marta Vilalta[1], Laura Castellini[1], Adam J. Krieg [ID] [9], Edward E. Graves [ID] [1], Laura D. Attardi[1] & Amato J. Giaccia [ID] [1,2✉]

Hypoxia plays a critical role in tumor progression including invasion and metastasis. To determine critical genes regulated by hypoxia that promote invasion and metastasis, we screen fifty hypoxia inducible genes for their effects on invasion. In this study, we identify v-maf musculoaponeurotic fibrosarcoma oncogene homolog F (MAFF) as a potent regulator of tumor invasion without affecting cell viability. MAFF expression is elevated in metastatic breast cancer patients and is specifically correlated with hypoxic tumors. Combined ChIP- and RNA-sequencing identifies IL11 as a direct transcriptional target of the heterodimer between MAFF and BACH1, which leads to activation of STAT3 signaling. Inhibition of IL11 results in similar levels of metastatic suppression as inhibition of MAFF. This study demonstrates the oncogenic role of MAFF as an activator of the IL11/STAT3 pathways in breast cancer.

[1] Division of Radiation and Cancer Biology, Department of Radiation Oncology, Stanford University, Stanford, CA, USA. [2] MRC Oxford Institute for Radiation Oncology, Department of Oncology, University of Oxford, Headington, UK. [3] Department of Biomedical Genetics, University of Rochester Medical Center, Rochester, NY, USA. [4] Department of Molecular Genetics and Microbiology, Duke University, Durham, NC, USA. [5] Department of Pathology, Department of Developmental Biology, Stanford University, Stanford, CA, USA. [6] Cell Cycle and Cancer Biology Research Program, Oklahoma Medical Research Foundation, Oklahoma, OK, USA. [7] Department of Radiation Oncology, Yonsei Cancer Center, Seoul, Republic of Korea. [8] Department of Chemical and Biomedical Engineering, Vanderbilt University, Nashville, TN, USA. [9] Department of Obstetrics and Gynecology, Oregon Health and Sciences University, Portland, OR, USA. ✉email: giaccia@stanford.edu

Metastasis, the dissemination of cancer cells to regional or distant sites, is the leading cause of cancer death. Metastasis requires sequential steps including invasion, intravasation, and extravasation, which are controlled by a variety of molecular mechanisms. Tumor hypoxia, which is caused by an imbalance between $O_2$ supply and consumption, results in reduced survival, resistance to radiotherapy and chemotherapy, metabolic reprogramming, increased angiogenesis and elevated invasion and metastatic potential[1–3]. Tumor cells adapt to hypoxia in large part by activating the HIF family of transcription factors, which are composed of α and β subunits. In contrast to the constitutively expressed HIFβ subunit, the HIFα subunit is regulated by various factors such as oxygen, growth factors, and free radicals. HIFα stabilization is largely regulated by three prolyl hydroxylases (PHDs), which require oxygen, 2-oxoglutarate, and iron to hydroxylate unique proline residues in the oxygen degradation domain (ODD) of the protein. Hydroxylation of HIFα enables the von Hippel-Lindau (VHL) protein complex to bind and initiate proteasomal degradation. A lack of oxygen, metabolic intermediates, or iron inhibits PHD activity and stabilizes the oxygen labile HIFα subunits. When stabilized, HIFα forms a heterodimer with HIFβ and binds to hypoxia response elements (HREs), resulting in transactivation of downstream genes that promote tumor progression[4,5]. In this study, we identified hypoxia-induced v-maf musculoaponeurotic fibrosarcoma oncogene homolog F (MAFF) as a HIF-1 target gene, which is regulated by hypoxia. Our data indicate that while MAFF does not affect primary tumor growth, it promotes disease progression by increasing the invasive and metastatic behavior of tumor cells.

MAF family proteins, first identified as viral oncogenes in chicken, are basic leucine zipper (bZIP) transcription factors, which can be divided into two groups: large MAFs and small MAFs[6]. Unlike large MAF proteins including MAFA (L-MAF), MAFB, c-MAF (MAF), and Neura Retinal (NRL), small MAF proteins, which consist of MAFF, MAFK, and MAFG, lack transactivation domains and form homodimers or heterodimers with the CNC family (NF-E2, NRF1, NRF2, NRF3) and BACH family proteins (BACH1 and BACH2)[7–9]. CNC and BACH family members require small MAF proteins to bind to the palindromic MAF-recognition element (MARE) (5′-TGCTGAC GTCAGCA-3′) or to the antioxidant response element (ARE), containing the core sequence, 5′-R(A/G)T(C/G)A(C/T/G) NNNGC-3′. While CNC family members including NRF2 positively transactivate target genes involved in antioxidative responses, BACH1 and BACH2, or homodimers of small MAF proteins can play a repressive role in regulating MARE and ARE-dependent genes by competing with NF-E2 and NRF2 for MARE or ARE binding[10,11]. Therefore, the functional role of small MAF proteins is highly dependent on their quantity and as well as the abundance of their binding partners.

Here, we demonstrate that MAFF, which is induced by HIF-1 under hypoxia, binds with BACH1 and promotes tumor invasion and metastasis by transcriptionally activating IL11 and promoting STAT3 pathways.

## Results

**Hypoxia-induced MAFF regulates tumor invasion and acts as a prognostic indicator**. To identify hypoxia-regulated genes involved in tumor progression, we took the top 50 hypoxia-induced genes that we previously identified (Supplementary Data 1)[12]. Analysis of TCGA human cancer data determined that these top 50 genes were genetically altered in most cancers but particularly in breast cancer patients, which had the highest frequency of gene amplification and copy number changes

(Fig. 1a)[13]. Using Gene Expression Omnibus (GEO) dataset (GSE19783), we found that the combined expression of these genes correlated with poor patient survival, suggesting an oncogenic role for them in breast cancer (Fig. 1b). From DAVID functional analysis, we found that these 50 genes had enriched GO terms for "glucose metabolic process", "regulation of apoptosis", and "positive regulation of cell migration" (Fig. 1c). Since tumor metastasis itself or the sequela of metastasis is the primary cause of patient death, we decided to focus on genes essential for invasion and migration. To more efficiently and specifically knockdown these 50 genes, we used endoribonuclease prepared siRNA (esiRNA) libraries and examined the impact of each gene on invasion under normoxia or hypoxia in MDA-MB-231 cells, a human breast cancer cell line with high metastatic potential. Cell invasion was quantified by combining data of "number of branches", "total length", and "total branching length" from cells growing on collagen I and normalized to cell survival (Fig. 1d and Supplementary Fig. 1a and b). We found that three genes (MAFF, LOX, AKAP12) significantly regulate cell invasion without having an effect on cell death when they were knocked down both under normoxia and hypoxia. While LOX and AKAP12 have been previously studied for their invasive role, we found that inhibition of MAFF consistently reduced cell invasion the most effectively in this 50 gene test set under normoxic and hypoxic conditions (Fig. 1d)[14,15].

To determine the clinical significance of MAFF in breast cancer patients, the expression of MAFF was examined in tissue microarrays. Analysis of staining intensity showed that MAFF expression was significantly higher in lymph node metastases when compared with the corresponding primary tumors, suggesting the role of MAFF in tumor progression and metastasis (Fig. 1e). We also found that there was a statistically significant positive correlation between hypoxia signatures and MAFF levels in breast cancer patients, supporting the hypoxic regulation of MAFF (Fig. 1f)[16]. Kaplan–Meier analysis further showed that high MAFF expression was significantly associated with reduced overall survival (OS) and metastasis-free survival (MFS) (Fig. 1g). These findings indicate that MAFF is strongly linked to hypoxia, tumor metastasis, and survival in breast cancer patients.

**MAFF is a transcriptional target of HIF1α**. Hypoxia-induced increases in MAFF levels were observed in a number of breast cancer cell lines by wetsern blotting. The specificity of MAFF antibody was confirmed by knocking down MAFF as well as other small MAF family members, MAFG and MAFK individually and then by testing the cross-reactivity of each small MAF antibody by Western blot analysis and qPCR (Supplementary Fig. 2a and b). We found higher MAFF expression in the more aggressive and metastatic cells (MDA-MB-231, Hs578T, MDA-MB-468) compared to non-invasive cells (MCF7, BT474) (Supplementary Fig. 2c). Expression of MAFF was significantly elevated under hypoxic conditions independent of variations in the normoxic expression of MAFF (Fig. 2a and b). Among MAFF binding partners, we observed that expression of NRF2, the most studied binding partner of small MAF proteins, was low in these breast cancer cell lines, suggesting that tumor invasion regulated by MAFF is NRF2 independent. In contrast, expression of BACH1 was high in all breast cancer cell lines.

In addition to hypoxia, we observed that treatment of cells with $CoCl_2$ and DMOG, chemical inducers of HIF activation, increased MAFF expression (Supplementary Fig. 2d). An increase in MAFF under hypoxia was also observed in ovarian, lung, liver, and brain tumor cell lines while MAFG and NRF2 expression was unchanged, indicating that hypoxia induction of MAFF is not cancer type specific but is mediated by a hypoxia-inducible factor

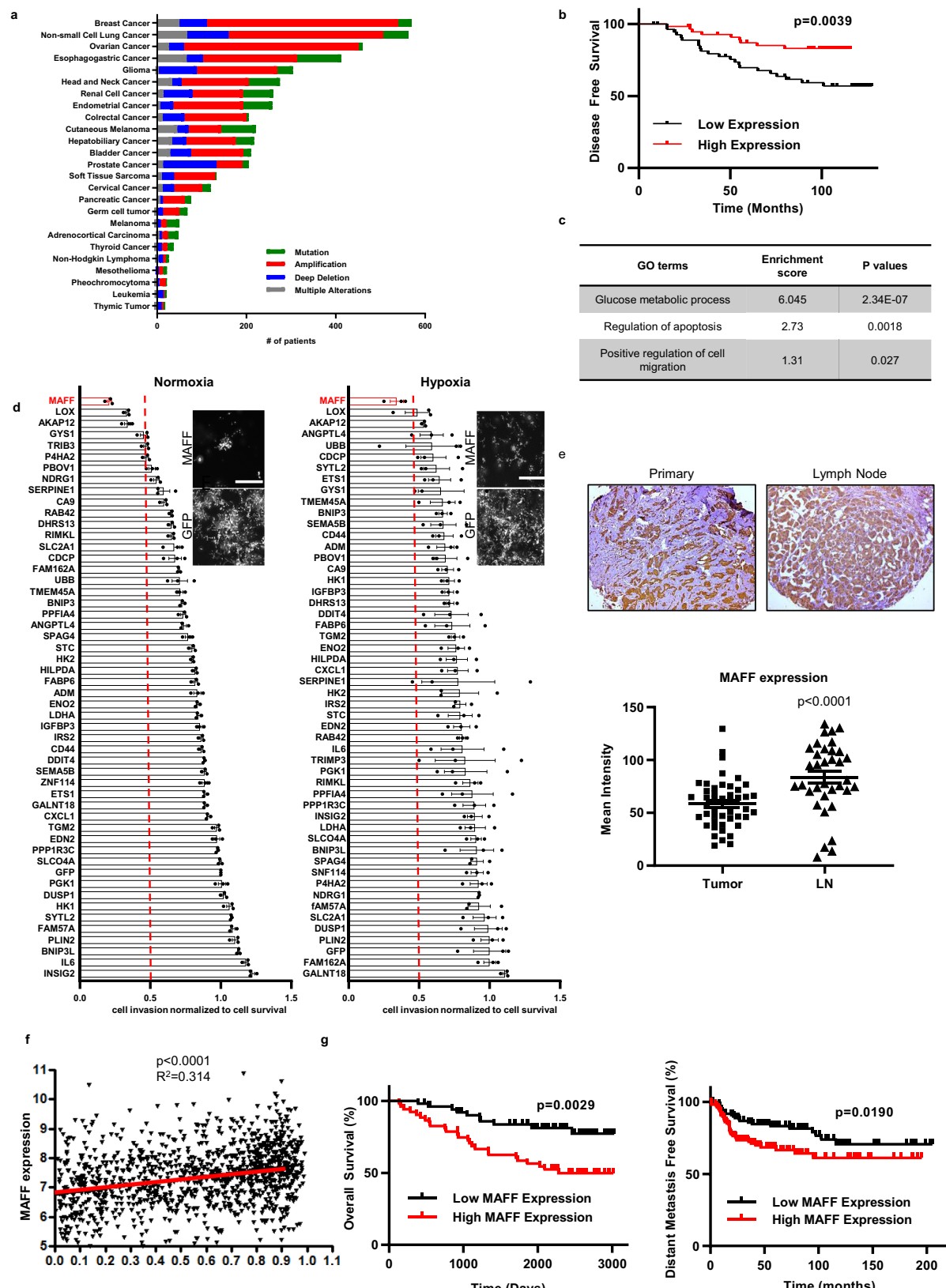

common to many cell lines such as HIF (Supplementary Fig. 2e and f).

To determine whether HIF-1, HIF-2, or both are required for hypoxia-mediated MAFF induction, we genetically silenced *HIF1A*, *EPAS1* (HIF2α), *ARNT* (HIFβ) using shRNA in MDA-MB-231 cells (Supplementary Fig. 2g and h). Inhibiting *HIF1A* or

*ARNT*, the constitutively active binding partner of HIF, in MDA-MB-231 cells greatly diminished hypoxia-induced MAFF expression (Fig. 2c and d). In contrast, inhibition of *EPAS* did not have a significant effect on the induction of MAFF under hypoxia (Supplementary Fig. 2h). These results indicate that MAFF is regulated under hypoxia in a HIF-1 dependent manner.

**Fig. 1 Hypoxia-induced MAFF regulates tumor invasion and acts as a prognostic indicator. a** 50 hypoxia-inducible genes were identified from RCC4-VHL, a clear-cell renal cell carcinoma line with reconstituted VHL, using microarray analysis[12]. TCGA data from cbioportal showed that these genes were altered in most of tumor cases, specifically in breast cancer. **b** Dataset from GEO showed that combined expression of 50 hypoxia genes were significantly correlated with overall survival in breast cancer patients (Log-rank test, $p = 0.0039$, GSE19783, $n = 56$ with low expression, $n = 56$ with high expression). **c** DAVID gene functional classification tool[20,21] classified 50 hypoxia-inducible genes into functional groups, which are correlated with "Glucose Metabolism", "Regulation of apoptosis", and "Positive regulation of cell migration". **d** RNAi screening was performed to determine genes regulating cell invasion using endoribonuclease prepared siRNA (esiRNA) libraries (MilliporeSigma). Collagen assay with tdTomato labeled MDA-MB-231 cells showed that MAFF regulates tumor cell invasion the most among 50 hypoxia-inducible genes (bar = 500μm) $n = 3$ biological replicates. One-Way ANOVA with multiple comparisons. $p < 0.0001$. **e** MAFF expression on breast cancer patient tissues was evaluated by staining tissue microarrays. When compared to primary tumor tissues using ImageJ, lymph node (LN) metastatic tissues showed higher intensity of MAFF expression ($n = 46$ for primary tissues, $n = 36$ for LN metastatic tissues). One-Way ANOVA with multiple comparisons. $p < 0.0001$. **f** While MAFF expression in breast cancer patients significantly correlated with hypoxia signatures (Log-rank test, $p < 0.0001$, $R^2 = 0.314$), **g** it was also a prognostic indicator for overall survival (OS) and distant metastasis-free survival (DMFS) in all or basal-like breast cancer patients (OS: GSE42568, $n = 52$ with low MAFF expression, $n = 52$ with high MAFF expression, $p = 0.0029$, DMFS: KM Plotter[63], $n = 117$ with low MAFF expression, $n = 115$ with high MAFF expression, $p = 0.0190$). Log-rank test. Graphs represent the mean per group and error bars represent the SEM.

To investigate whether HIF1 regulates *MAFF* transcription through direct binding, we searched for consensus HRE sequences, 5′-(A/G)CGTG-3′ spanning between −5 kb and +1 kb of the transcription start site of *MAFF*[17]. Using anti-HIF-1α antibody, DNA fragments that bind to HIF-1 were immunoprecipitated and analyzed by qRT-PCR. Under hypoxia, two HRE sites, HRE8 and HRE9 had robust enrichment of HIF-1α binding compared to IgG controls, suggesting that *MAFF* is a direct target of HIF-1 (Fig. 2e). Primers flanking the HRE site in *JMJD1A* were used as a positive control[12]. To further investigate whether HIF-1 transactivates gene expression through the HRE8 or HRE9 sequences of *MAFF*, we generated reporter gene constructs by subcloning these sequences into the pGL4-luciferase reporter vector (Promega). As a negative control, we added mutations in the core HRE sequences. A pGL4-HRE-luciferase reporter containing four HRE sequences was used as a positive control. Under hypoxia, luciferase activity of the reporter gene under the control of HRE8 or HRE9 was significantly increased (Fig. 2f). In contrast, reporter genes with mutated constructs of HRE8-mut and HRE9-mut did not respond to low oxygen tensions, indicating that these sequences are required to transcriptionally activate *MAFF* under hypoxia.

**MAFF regulates tumor cell invasion in vitro.** To confirm our findings from esiRNA screening, we silenced *MAFF* using shRNA in MDA-MB-231 cells (Fig. 3a and Supplementary Fig. 3a). In MDA-MB-231 cells, *MAFF* knockdown did not alter in vitro cell growth under normoxia or hypoxia, indicating that MAFF does not play a key role in primary tumor growth (Fig. 3b). To determine whether MAFF inhibition altered tumor metabolism, we measured lactate and pyruvate levels from the cell media. While hypoxia treatment increased lactate production, MAFF knockdown did not significantly change levels of lactate either under normoxia or hypoxia (Supplementary Fig. 3c). Using the Seahorse XF analyzer, we also confirmed that extracellular acidification rates (ECAR) did not change when MAFF was inhibited (Supplementary Fig. 3d). Interestingly, we observed that pyruvate levels were significantly decreased in the absence of MAFF (Supplementary Fig. 3c), suggesting that MAFF might play a role in oxidative phosphorylation but not in glycolysis.

As we observed from esiRNA screening (Fig. 1d), knocking down MAFF significantly reduced tumor cell invasion and migration through type I collagen and Matrigel-coated transwell membranes (Fig. 3c). When MAFF was re-expressed with wobble mutations in these knockdown cells, inhibition of tumor cell invasion was rescued (Fig. 3d).

Similar results were observed when MAFF was overexpressed in MCF7, indicating that MAFF expression is not critical for cell proliferation (Fig. 3d and e, Supplementary Fig. 3b). Interestingly,

typically non-invasive MCF7 cells exhibited enhanced cell invasion when MAFF was overexpressed (Fig. 3e and f).

To evaluate whether these observations resulted from the absence of NRF2, we also tested ovarian and lung cancer cells with high NRF2 expression. In OVCAR8, an invasive ovarian cancer cell line with high NRF2 expression, loss of MAFF reduced cell invasion (Supplementary Fig. 3e and g). In addition, overexpression of MAFF in A549 human lung cancer cells, which exhibit a non-invasive phenotype, displayed increased tumor cell invasion and migration (Supplementary Fig. 3f and h). In these cells NRF2 expression was not altered by hypoxia. Our results suggest that high MAFF levels enhance tumor cell invasion regardless of NRF2 status.

**MAFF regulates tumor metastasis in vivo.** To investigate the role of MAFF in metastasis in vivo, we used orthotopic mouse tumor models. MDA-MB-231 cells were injected into the mammary fat pad of nude mice to measure the effect of MAFF inhibition on primary tumor growth. Compared to the control group, the MAFF deficient tumor group did not show any significant differences in primary tumor growth, which was consistent with in vitro data on MAFF not playing a significant role on growth and cell survival (Fig. 4a). Tumor cell proliferation as determined by Ki67 staining was also not affected by *MAFF* knockdown (Fig. 4b). Interestingly, tumor microvessel density (MVD) measured by MECA32 staining was significantly decreased in tumors with *MAFF* knockdown, suggesting that MAFF also affects tumor angiogenesis, which is required for tumor metastasis (Fig. 4b)[18]. Overlapping regions of MAFF and hypoxic areas, which were identified using the hypoxia marker pimonidazole were frequently observed (~87% colocalization), supporting a link between hypoxia and MAFF expression (Supplementary Fig. 4a).

In contrast to orthotopic tumor growth, lung colonization after tail vein injection was largely reduced when MAFF was knocked down in MDA-MB-231 cells (Fig. 4c). Metastatic tumor burden in the lung was quantified by histological analysis of lung nodules and PCR based measurements of human *GAPDH* expression in the mouse lung. These results indicate that inhibition of MAFF results in decreased metastatic foci.

The metastatic role of MAFF was further investigated using a second metastatic tumor model. Following previously described procedures, OVCAR8 cells, which express high NRF2, were intraperitoneally injected into nude mice and allowed to form metastatic foci[19]. Consistent with the effects on MDA-MB-231 cells, sh*MAFF* also resulted in decreased metastatic burden, indicating that MAFF regulates tumor metastasis in a number of tumor types (Fig. 4d).

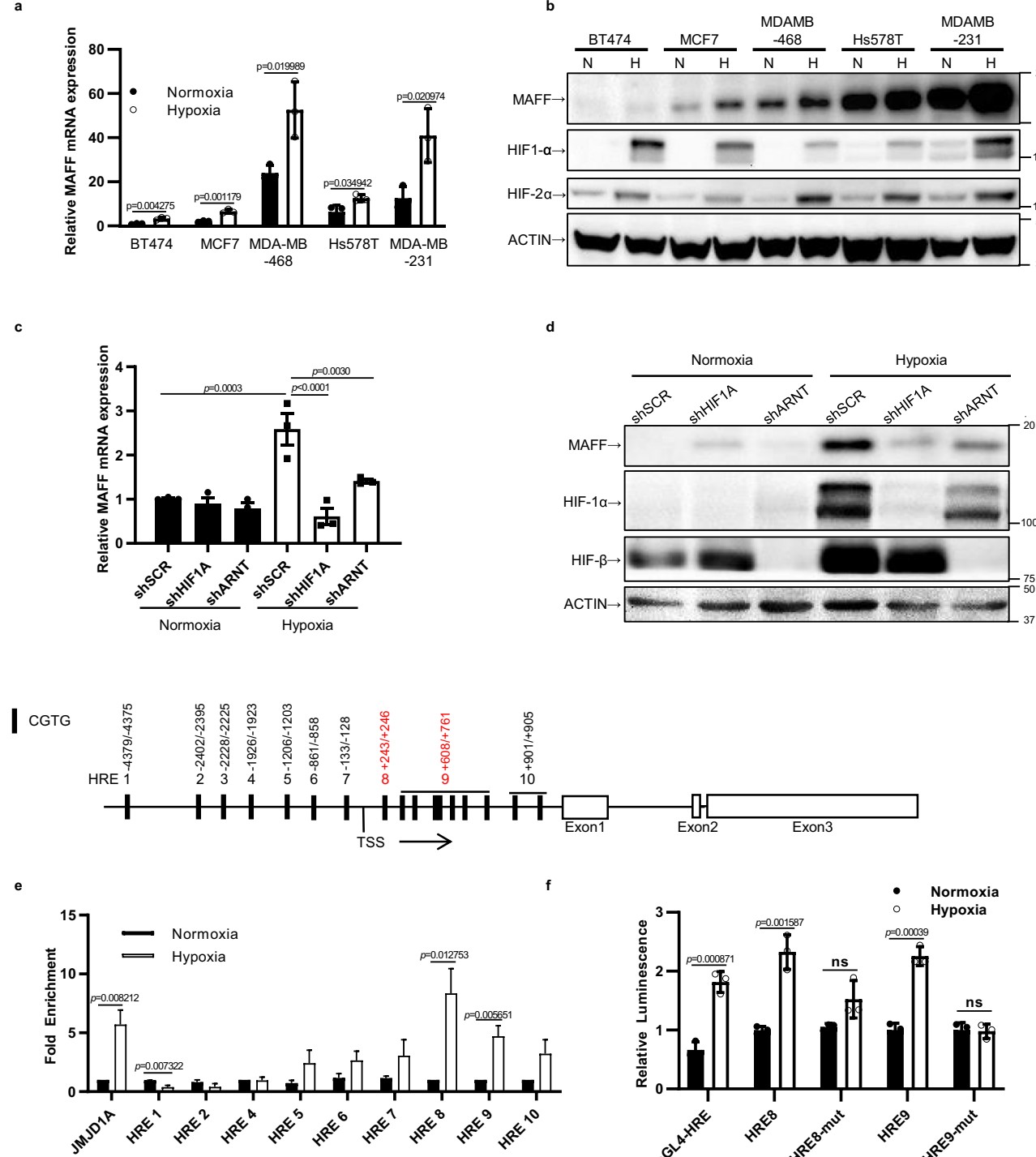

**Fig. 2 MAFF induction under hypoxia is regulated by HIF-1. a** *MAFF* mRNA expression was elevated under hypoxia in breast cancer cell lines. $n = 3$ biological replicates. Unpaired $t$-test. **b** While basal expression of MAFF was different, hypoxia increased MAFF expression in all breast cancer cell lines. **c**, **d** Knocking down *HIF1A* or *ARNT* (HIFβ) prevented MAFF mRNA and protein induction under hypoxia. One-Way ANOVA with multiple comparisons. **e** ChIP assay was performed to identify a HIF-1 binding site on the *MAFF* promoter (HRE8 and 9) (TSS: transcription start site). $n = 4$ biological replicates. Unpaired $t$-test. **f** Luciferase reporter assays showed that transcription activity of HRE8 or HRE9 was increased under hypoxia compared to mutated HRE8 or HRE9. pGL4-HRE, which contains 4x HRE regions, was used as a positive control. $n = 3$ biological replicates. Unpaired $t$-test. Graphs represent the mean per group and error bars represent the SEM. ns: not significant.

**Genome-wide analysis of MAFF-mediated gene regulation by RNA- and ChIP-sequencing.** To identify genome-wide downstream targets of MAFF in tumor cells that are involved in tumor invasion and metastasis, we performed high throughput RNA-sequencing of MDA-MB-231 cells exposed to normoxia or

hypoxia for 24 hours, with or without genetic inhibition of *MAFF* by siRNA. When MAFF was inhibited under normoxia, 415 genes were identified to be significantly changed, while hypoxia altered 240 genes. Among these genes, 106 genes were regulated both under normoxia and hypoxia. To obtain a global overview of the

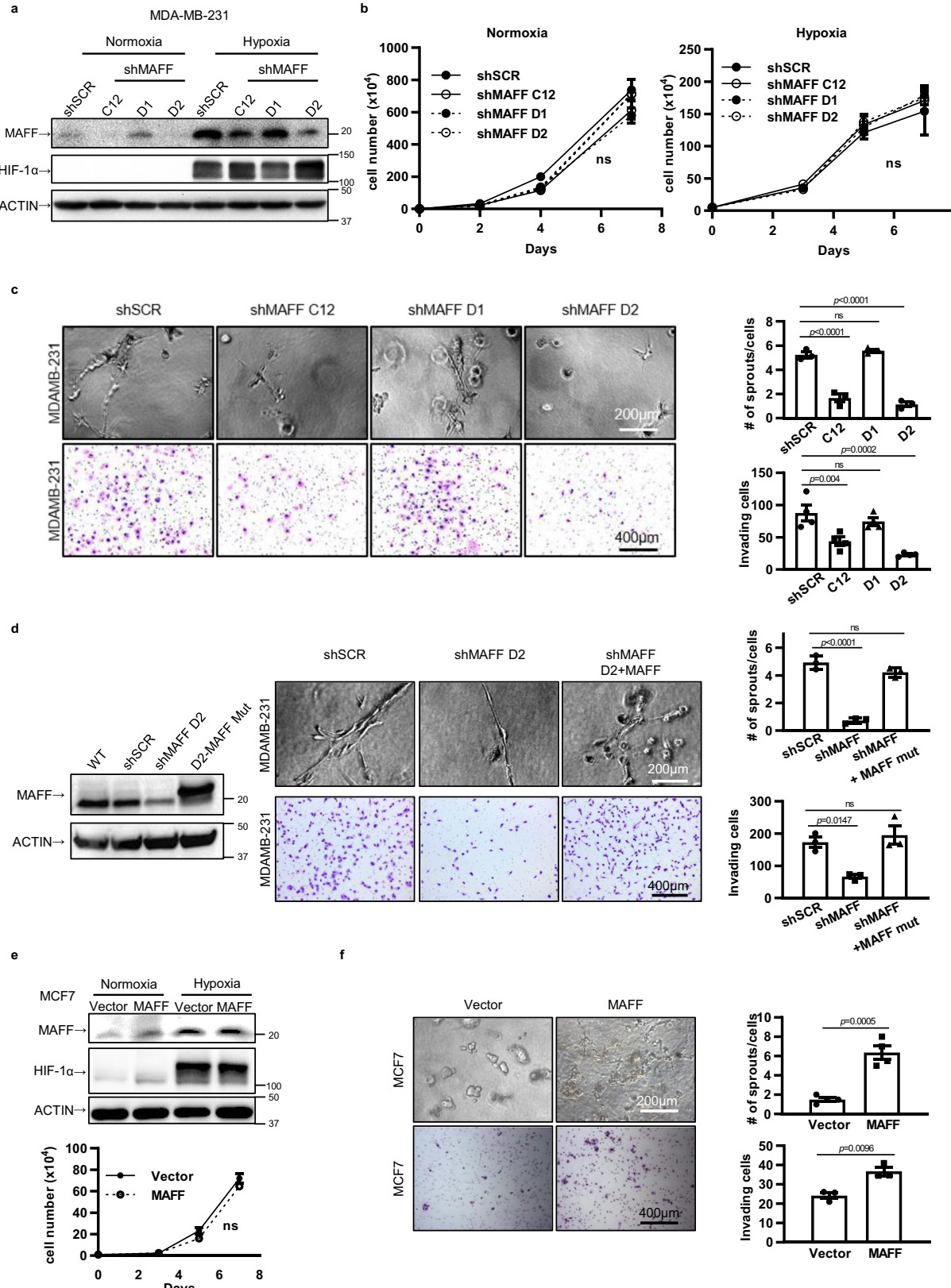

biological functions of proteins encoded by genes identified from RNA-sequencing under normoxia or hypoxia or both conditions, we performed DAVID analysis to determine KEGG pathways and Gene Ontology (GO) terms[20] and found that genes involved in pathways of "Blood vessel development", "TGFβ signaling", and "Regulation of cell adhesion/morphogenesis" were significantly changed in the absence of MAFF expression (Fig. 5c). These observations suggest MAFF-mediated gene activation and repression while supporting our findings on the role of MAFF in tumor cell invasion and metastasis. GSEA data additionally showed that metastasis and migration-related genes were altered by MAFF (Fig. 5d).

**Fig. 3 MAFF regulates tumor cell invasion. a** *MAFF* was knocked down using three different clones of shRNA against *MAFF*. Western blot data showed that while two of clones (C12 and D2) decreased MAFF expression significantly, one clone had little effect on MAFF expression (D1), which we decided to use as a negative control. **b** MAFF inhibition using shRNA did not affect tumor cell growth in MDA-MB-231 cells both under normoxia and hypoxia. $n = 3$ biological replicates. Unpaired *t*-test. **c** MAFF knockdown in MDA-MB-231 cells decreased tumor cell invasion through collagen (upper panels, $n = 3$ biological replicates) or Matrigel-coated transwell membranes (lower panels, $n = 4$ biological replicates). Quantification is included in graphs. One-Way ANOVA with multiple comparisons. **d** *MAFF* knockdown was rescued by overexpressing MAFF cDNA with mutations in the shRNA targeting region. Decreased tumor cell invasion by MAFF inhibition was reactivated when MAFF was re-expressed (mut: mutant). Tumor cell invasion was determined using collagen assay (upper panels) or Matrigel-coated transwell membrane assay (lower panels). $n = 3$ biological replicates. One-Way ANOVA with multiple comparisons. **e** MAFF overexpression in MCF7 cells did not show any significant difference in cell growth rates (ns: not significant). $n = 3$ biological replicates. Unpaired *t*-test. **f** Increased invasive phenotypes were observed in MAFF overexpressing MCF7 using collagen assay (upper panels, $n = 3$ biological replicates) and transwell invasion assay (lower panels, $n = 4$ biological replicates). Unpaired *t*-test. (Graphs represent the mean per group and error bars represent the SEM. ns: not significant).

To further identify genes that are directly regulated by MAFF through transactivation, we performed ChIP-sequencing on MAFF bound DNA from MDA-MB-231 cells with shRNA inhibited endogenous MAFF and stably overexpressing V5-tagged MAFF with a wobble mutation. Global analysis of ChIP-sequencing data indicates that 40.44% of MAFF binding is in the promoter region and 27.36% is in intron regions (Fig. 5e). From MEME analysis, a MAFF enriched motif was found that is similar with the consensus MARE sequence (Fig. 5f). By combining profiles from RNA-sequencing and ChIP-sequencing, we highlighted 358 genes under normoxia, 136 genes under hypoxia, and 66 genes under both conditions that were directly bound and regulated by MAFF (Fig. 5g, Supplementary Data 2). We further identified MAFF-regulated genes that were specifically involved in tumor cell invasion and metastasis using DAVID and Pubmatrix, a text mining tool (Supplementary Fig. 5a)[21,22]. Using these tools, we found three genes (*TGFB2, BMP4, COL1A1*) that were upregulated and two genes (*IL11, TNC*) that were downregulated when MAFF was inhibited under both normoxic and hypoxic conditions. RNA-sequencing results were validated by qRT-PCR, and only *IL11* and *TGFB2* showed similar changes by *MAFF* knockdown and hypoxia (Supplementary Fig. 5b). In OVCAR8 cells, which highly express NRF2, loss of MAFF also resulted in inhibition of *IL11*, while the rest of these genes were not significantly changed under hypoxic conditions by the loss of MAFF (Supplementary Fig. 5c).

We further searched the MAFF binding peaks in target gene promoters using the UCSC genome browser (Fig. 5g). Although expression of *HMOX1* was not significantly altered by MAFF knockdown, we found binding peaks on two known *HMOX1* promoter regions[23]. Likewise, we identified a strong peak on promoter region of IL11 possessing MARE/ARE sequence (GCGAGCTCA), indicating that MAFF directly regulates IL11 transcription.

Since *IL11* appears to be a direct target of MAFF while other gene regulation seems to be more cell type specific, we decided to focus on the role of MAFF-mediated *IL11* expression in tumor cell invasion.

IL11, which is a member of the IL6 family, is a cytokine involved in inflammation, immune responses, and bone resorption[24]. Recent studies further revealed that IL11 plays a role in tumorigenesis by altering angiogenesis and metastasis[25,26]. To determine whether IL11 is responsible for MAFF-mediated tumor cell invasion, *IL11* was silenced in MDA-MB-231 cells using shRNA (Supplementary Fig. 6a). *IL11* knockdown significantly reduced invasion (Fig. 6a). In contrast, MDA-MB-231 cells increased invasiveness with IL11 overexpression (Fig. 6b, and Supplementary Fig. 6b). In addition, exogenously expressed IL11 rescued the decreased invasion in MAFF knockdown cells, suggesting that MAFF regulates tumor cell invasion through the IL11 pathway. Similar results were observed when IL11 was expressed in MCF7 cells (Supplementary Fig. 6c–e).

Cellular responses of IL11 are mediated by STAT3 activation, which is known to be associated with tumorigenesis and metastasis[27]. To determine whether MAFF-induced IL11 affects tumor cell invasion through STAT3 signaling, we first measured phospho-STAT3 protein levels. As previously reported, we also found that phospho-STAT3 was increased under hypoxia (Fig. 6c)[28].

Furthermore, *MAFF* knockdown or *IL11* knockdown in MDA-MB-231 cells suppressed phospho-STAT3 induction, while overexpression of IL11 protein in *MAFF* knockdown cells rescued STAT3 activation (Fig. 6d, e). In addition, when MAFF was overexpressed in MCF7 cells, IL11 and phospho-STAT3 were increased even under normoxia (Supplementary Fig. 6f). These results indicate that MAFF can activate the STAT3 pathway by directly regulating IL11.

While RNA-sequencing data indicated that MAFF is involved in blood vessel formation, STAT3 is also known to play a role in angiogenesis[29]. To determine whether MAFF regulates angiogenesis, which could enhance tumor invasion and metastasis, tube formation assay was performed (Fig. 6f). When a GFP-labeled human umbilical vein endothelial cell line (HUVEC) was incubated with cell media from MDA-MB-231 cells deficient in *IL11* through *IL11* Cas9-CRISPR treatment, tube formation was significantly reduced. We also observed that cells stably expressing *MAFF* shRNA decreased angiogenic features while expression of IL11 rescued the reduction (Fig. 6g). As previously reported[30], knocking down STAT3 using siRNA decreased tumor cell invasion and tube formation of HUVEC cells (Supplementary Fig. 6g and h), indicating MAFF regulates angiogenesis through an IL11/STAT3 pathway. These results further support our previous in vivo data showing decreased MVD in MAFF knockdown tumors (Fig. 4b).

**MAFF regulates tumor cell invasion and metastasis through a BACH1-dependent pathway.** We hypothesized that MAFF-mediated transactivation or repression depends on its binding partner. To identify binding partners of MAFF involved in tumor cell invasion and metastasis, we first performed affinity purification and mass spectrometry (AP-MS). Protein complexes of MAFF and its binding partners were then isolated from cell lysates by co-immunoprecipitation, and then analyzed by mass spectrometry[31]. Among known interactors of MAFF, we found that BACH1 was the only binding partner of MAFF in MDA-MB-231 cells (Fig. 7a). Co-immunoprecipitation analysis followed by immunoblotting further indicated that BACH1 binds MAFF, which increased further under hypoxia while BACH1 expression was unchanged (Fig. 7b). BACH1, a distantly related CNC protein, plays a repressive role in regulating antioxidant genes by competing with CNC proteins for MARE or ARE binding[32]. Interestingly, previous studies revealed that BACH1 promotes tumor invasion and metastasis while not affecting primary tumor

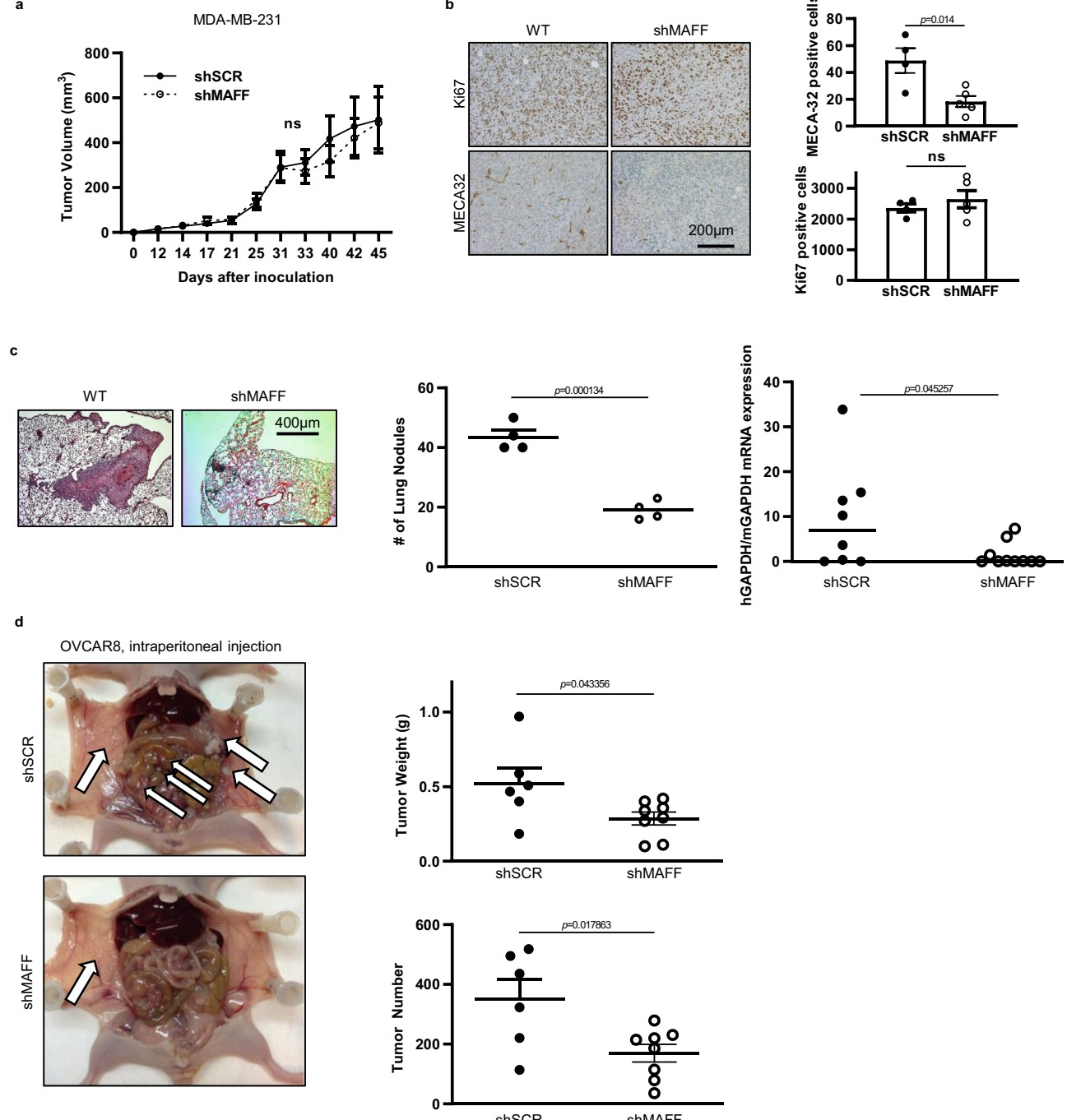

**Fig. 4 MAFF regulates in vivo tumor metastasis.** MDA-MB-231 cells were orthotopically injected into mammary fat pads of nude mice ($10^6$ cells/50 μl/ mouse) to determine the role of MAFF in tumor growth. **a** MAFF inhibition did not affect tumor growth in vivo (ns: not significant). $n = 9$ shSCR, $n = 7$ shMAFF. Two-Way ANOVA. **b** While cell proliferation (Ki67 positive staining) was not changed, microvessel density (MECA32 positive staining) significantly decreased when MAFF was inhibited ($n = 4$ shSCR, $n = 5$ shMAFF). Unpaired $t$-test. **c** Lung colonization of tumor cells was evaluated either by H&E staining ($n = 4$) or qPCR to measure human *GAPDH* expression ($n = 9$ shSCR, $n = 7$ shMAFF). Unlike primary tumor growth, *MAFF* knockdown reduced lung metastasis. Unpaired $t$-test. **d** Intraperitoneal injection of OVCAR8 cells also showed that inhibition of MAFF expression was significantly correlated with reduced tumor metastasis. Arrow indicates tumor nodules formed in intraperitoneal cavity. $n = 6$ shSCR, $n = 8$ shMAFF. Unpaired $t$-test. Graphs represent the mean per group and error bars represent the SEM. ns: not significant.

growth[33–35]. To confirm previous findings as well as to determine the role of BACH1 as a MAFF binding partner, we knocked down BACH1 and measured tumor cell growth (Supplementary Fig 7a). The effect of IL11 or STAT3 on cell growth was also determined after silencing these genes using a Cas9-CRISPR system. Consistent with MAFF knockdown, we observed that inhibition of

BACH1, IL11, or STAT3 did not change primary tumor growth, suggesting that their main role in cancer was not the regulation of cell proliferation in our experimental settings. We next investigated whether BACH1 regulates IL11 expression and invasion in conjunction with MAFF. *BACH1* inhibition by shRNA in MDA-MB-231 cells downregulated *IL11* mRNA levels as well as

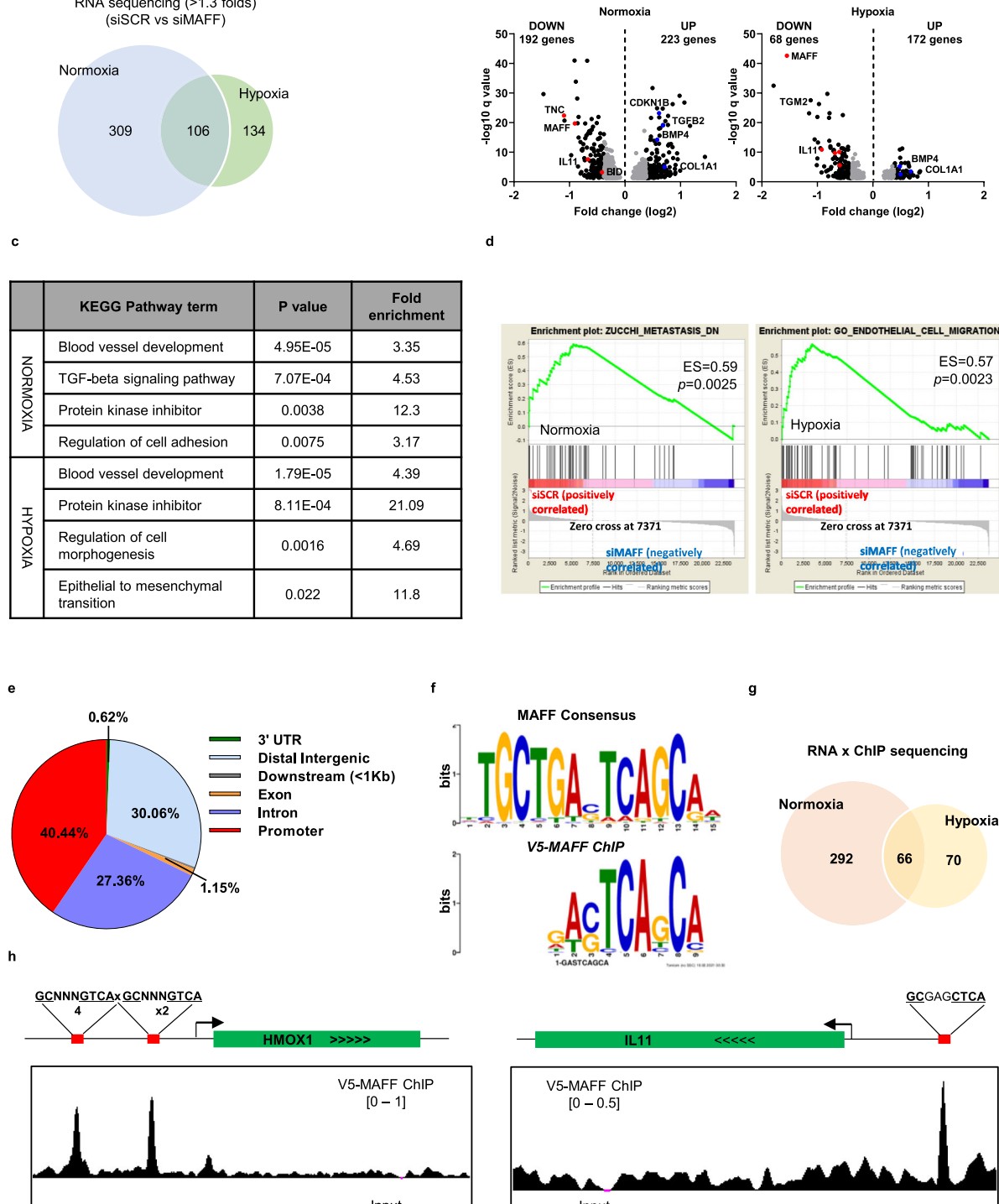

phospho-STAT3 expression both under normoxia and hypoxia (Fig. 7c and d). To further determine the direct binding of MAFF and BACH1 on IL11 promoter, we performed ChIP and luciferase reporter assays. Using ChIP-qPCR with primers spanning the identified MARE/ARE region, we determined that MAFF and BACH1 exhibited increased binding to the *IL11* promoter under

hypoxia (Fig. 7e). Using this same region subcloned into pGL4 luciferase, we found that the expression of *IL11* was increased under hypoxia and inhibited under both normoxia and hypoxia (Fig. 7f). When BACH1 or both BACH1 and MAFF were inhibited, tumor cell invasion and tube formation of HUVEC cells were also significantly decreased (Fig. 7g). The decreases in

**Fig. 5 MAFF target gene identification by genome-wide analysis.** MDA-MB-231 cells were treated under normoxia or hypoxia with or without siMAFF for 24 h. MAFF target genes were determined by analyzing RNA-sequencing and ChIP-sequencing data. **a** When *MAFF* was knocked down, 415 or 240 genes were significantly altered under normoxia or hypoxia, respectively and among them 106 genes were affected both under normoxia and hypoxia. **b** Volcano plot showed that 192 genes were downregulated and 223 genes were upregulated under normoxia. Under hypoxia, 68 genes were downregulated and 172 genes were upregulated. **c** DAVID[20,21] analysis determined that genes involved in pathways of "Blood vessel development", "TGFβ signaling", and "Regulation of cell adhesion/morphogenesis" were significantly altered in the absence of MAFF expression. **d** GSEA analysis revealed that genes involved in metastasis (normoxia) or migration (hypoxia) were regulated by MAFF expression. **e** The genomic location of MAFF binding sites was identified by *cis*-regulatory element annotation system (CEAS). **f** A MAFF binding motif based on MARE was identified using the motif discovery algorithm, Multiple Expectation Maximizations for Motif Elicitation (MEME). **g** When we combined RNA-sequencing and ChIP-sequencing, 292 genes were regulated under normoxia and 70 genes under hypoxia. 66 genes were regulated both under normoxia and hypoxia. **h** ChIP-sequencing data showed enriched DNA sequence tags, which were highlighted in red using the UCSC genome browser. While *HMOX1* showed two peaks from previously known MAFF binding sites, MAFF binding peak was found in *IL11* promoter, which included the MARE/ARE sequence (GCGAGCTCA).

invasion and tube formation were rescued by IL11 over-expression. These results suggest that IL11 is a direct target of MAFF and BACH1 and regulates cell invasion and angiogenesis. In contrast to BACH1, NRF2 knockdown in MDA-MB-231 did not affect IL11 expression or tumor cell invasion, since it is already low in expression (Supplementary Fig. 7b).

**MAFF enhances tumor metastasis through IL11/STAT3 pathways in vivo and their combined expression predicts patient survival.** To determine if MAFF and its downstream targets, IL11/STAT3, play a significant role in tumor metastasis, we performed in vivo experiments using MDA-MB-231 cells injected into the mammary fat pad of NSG mice. This model is known to form spontaneous lung metastasis within two months[36]. We compared the formation of metastasis in lungs by MDA-MB-231 cells stably expressing shSCR, sh*MAFF*, or sh*MAFF* with IL11 overexpression. Consistent with our previous in vivo studies using athymic nude mice (Fig. 3a), knocking down MAFF did not change primary tumor growth rates. However, the mouse group with sh*MAFF* and IL11 overexpression showed increased tumor growth, which resulted from elevated expression of IL11 (Supplementary Fig. 8a and b). In these experiments, mice were sacrificed when their primary tumors reached a similar median size (~1000 mm$^3$). After sacrifice, mouse lungs from the different groups were excised to quantify the number of metastatic nodules. MAFF inhibition significantly decreased metastasis from the primary tumor to the lung (Fig. 8a). However, compared to shSCR group, MAFF knockdown MDA-MB-231 tumors with IL11 overexpression displayed similar levels of tumor metastasis as shSCR controls, mirroring the rescue effects from our in vitro studies. The elevated levels of IL11 were confirmed from mouse serum using a human IL11 ELISA kit (Fig. 8b). In contrast, IL6, which is a known regulator of the STAT3 pathway[37], was not significantly altered by MAFF in mouse serum. Ki67 staining of primary tissues was not significantly different between groups, indicating that tumor proliferation was similar at the time of sacrifice (Supplementary Fig. 8c). Expression of phospho-STAT3 and MECA32 staining was also examined from primary tumor tissues to examine the effect of MAFF and IL11 on STAT3 pathways and microvessel density (Fig. 8c and d). As we observed from our in vitro studies, while the inhibition of MAFF decreased expression of phospho-STAT3 and MECA32, IL11 over-expression rescued these inhibitory effects. Tumor tissues exhibited significant colocalization of MAFF and HIF-1 (66% colocalization), MAFF and IL11 (94% colocalization), and BACH1 and IL11 (75% colocalization), suggesting their inter-active role in cancer (Supplementary Fig. 8d).

Finally, survival analysis of breast cancer patients indicated that expression of both MAFF and BACH1 or MAFF, BACH1, and IL11 was significantly correlated with distant metastasis-free survival, further strengthening the prognostic significance of MAFF/IL11

pathways (Fig. 8e). Taken together, these data demonstrate that MAFF is a positive regulator of tumor cell invasion and metastasis by activating IL11/STAT3 pathways when binding to BACH1, especially under hypoxic conditions (Fig. 9).

## Discussion

Tissue hypoxia in solid tumors leads to the activation of HIF pathways which affect tumor angiogenesis, metabolism, and metastatic potential[38]. In this study, we describe 50 hypoxia-induced genes, found from microarray analysis[12], that are increased in most tumor types and are highly correlated with patient survival particularly in breast cancer (Fig. 1 and Supplementary Fig. 1). Pathway analysis confirmed that these hypoxia-induced genes are involved in the regulation of cancer cell metabolism and migration, indicating their significant role in tumor progression. We previously found that the hypoxia-inducible genes *LOX* and *AKAP12* played critical roles in tumor invasion and metastasis in breast cancer or melanoma[14,15]. Using esiRNAs to specifically inhibit each of these fifty genes we identified *MAFF* as a hypoxia-induced gene, which had the most dramatic effect on tumor cell invasion when inhibited. In contrast to LOX and AKAP12, we found that MAFF transcriptionally regulates a different set of target genes to promote tumor invasion and metastasis.

Small MAF proteins have been mainly known for promoting antioxidant responses as binding partners of CNC family members and BACH proteins[7]. However, the individual roles of small MAF proteins are poorly understood. Recent studies provide significant insights into NRF2 and BACH1 in tumor progression, including angiogenesis, tumor metabolism, endothelial-to-mesenchymal transition (EMT), and metastasis[33–35,39–47], suggesting possible links between small MAFs and tumor progression. Here, we provide strong evidence that the small MAF protein MAFF is a key regulator of tumor invasion and metastasis, especially under hypoxic conditions.

Our genome-wide analysis using RNA- and ChIP-sequencing determined *IL11* as a direct target of endogenous MAFF, which is responsible for an invasive tumor phenotype (Fig. 5 and Supplementary Fig. 5). The role of IL11 in tumor metastasis was first studied in bone metastatic breast cancer due to its osteolytic properties[26]. Further studies showed that IL11 is highly elevated in patient tissues with a variety of cancers[26,27,48,49]. Its role is not limited to bone metastasis only, but to tumorigenesis and tumor metastasis in general[27,50]. Expression of phospho-STAT3 (Tyr705) indicates that MAFF also activates the STAT3 path-way, a downstream target of IL11. We observed that STAT3 phosphorylation is downregulated when MAFF or IL11 were lost and upregulated when MAFF or IL11 were overexpressed (Fig. 6). Under hypoxia, we also found that induction of IL11 and STAT3 activation is inhibited by MAFF knockdown. While IL11 and STAT3 are potential therapeutic targets, our data provide insights on how hypoxia regulates IL11 and STAT3 in cancer[28,51,52].

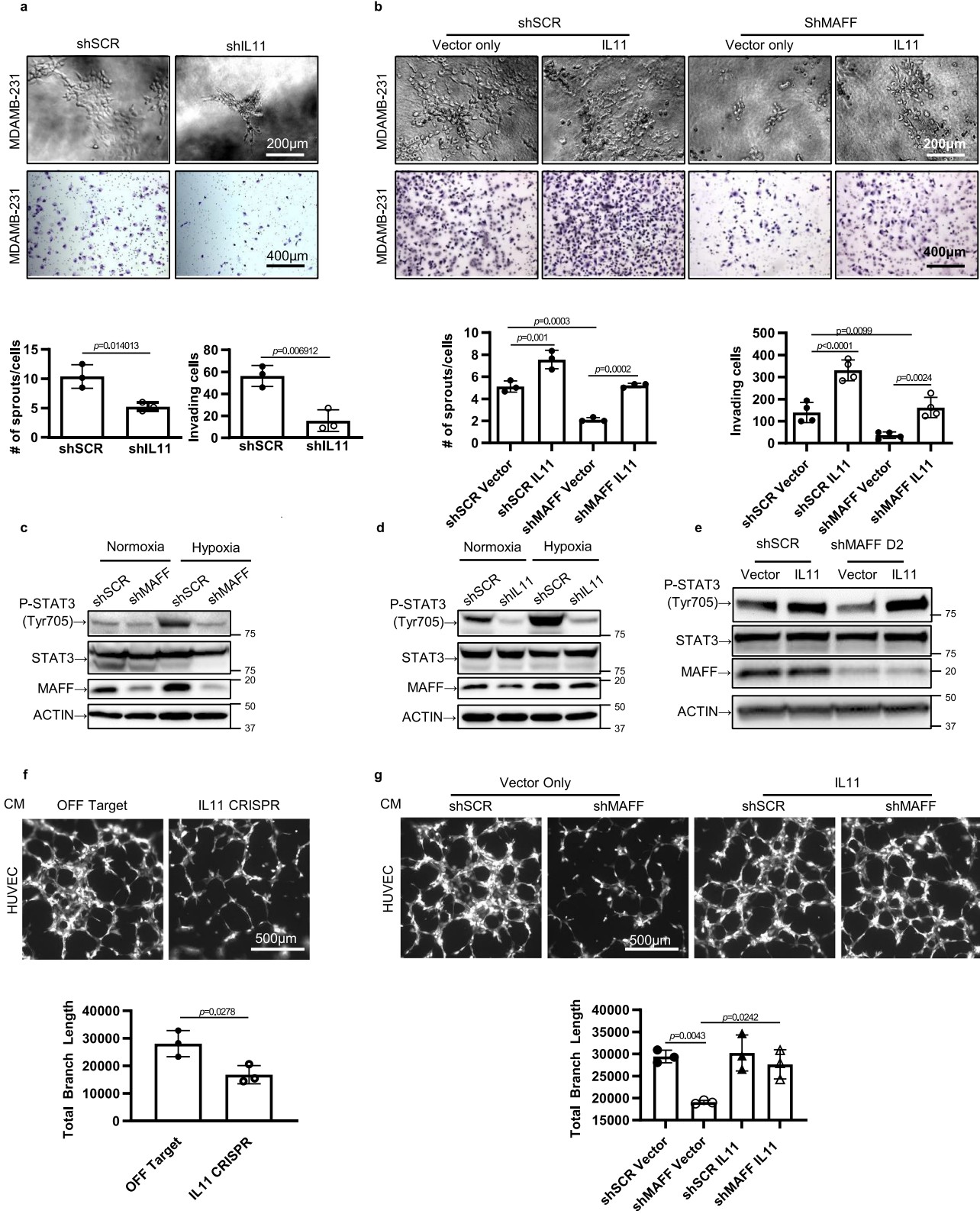

In our in vivo studies, we also show that *MAFF* knockdown decreases tumor lung metastasis in both tail vein injection and spontaneous metastasis models (Figs. 4 and 8). In the orthotopic mouse model, we also confirm that overexpression of IL11 restores the levels of tumor metastasis and expression of phospho-STAT3 expression in the absence of MAFF, supporting a critical role of IL11 in MAFF-mediated tumor metastasis

(Fig. 9). Although we did not observe changes in human IL6 from the serum of mice harboring MDA-MB-231 tumor cells with or without MAFF expression, we still need to consider the effect of IL6 pathways through trans-signaling. While the classic IL6 pathway is activated through the binding of IL6 to IL6 receptor (IL6R) on the cell surface, proteolytic cleavage of IL6R by ADAM17 or ADAM10 or alternative splicing of IL6R mRNA

**Fig. 6 MAFF-mediated IL11 regulates tumor cell invasion. a** Knocking down *IL11* significantly decreased invasive phenotypes of MDA-MB-231 through collagen (upper panels) and Matrigel-coated transwell membranes (lower panels). $n = 3$ biological replicates, Unpaired *t*-test. **b** The rescue effect of IL11 expression on decreased tumor cell invasion was evaluated in MDA-MB-231 cells with or without *MAFF* knockdown. $n = 3$ (collagen assay, upper panels) or 4 (transwell membrane assay, lower panels) biological replicates. One-Way ANOVA with multiple comparisons. **c** STAT3 activation was evaluated in MDA-MB-231 cells treated under normoxia or hypoxia by Western blotting. Hypoxia increased phospho-STAT3 (Tyr705), while MAFF knockdown inhibited its induction both under normoxia and hypoxia. **d, e** The downstream effect of IL11 on STAT3 activation was determined after *IL11* was knocked down (**d**) or overexpressed with or without MAFF inhibition (**e**). Phospho-STAT3 expression was investigated by Western blotting. **f, g** Endothelial tube formation assay was performed using GFP-labeled HUVEC cells. Cell culture media were transferred from MDA-MB-231 cells infected with IL11 Cas9-CRISPR (**f**) or *MAFF* shRNA with or without IL11 overexpression (**g**). $n = 3$ biological replicates, Unpaired *t*-test (**f**) or One-Way ANOVA with multiple comparisons (**g**). Graphs represent the mean per group and error bars represent the SEM.

results in trans-signaling of IL6 through soluble IL6R (sIL6R)[53]. The interaction of sIL6R and gp130 is able to then activate STAT3 pathways in cells that do not express IL6R. Since ADAM10 and ADAM17 are hypoxia targets, we cannot rule out the possible involvement of sIL6R in elevated STAT3 signaling[54,55]. Also, crosstalk between human and murine IL6 pathways in the murine tumor microenvironment by tumor-infiltrating immune cells or fibroblasts can potentially prime the metastatic niche and hyperactivate STAT3 pathways in cancer cells through IL6 trans-signaling pathways[56,57]. Therefore, further studies are required to more rigorously determine the role of IL6 in HIF-1-MAFF-STAT3 pathways.

Although NRF2 has been known for its role in tumor initiation, recent studies emphasize its effect on tumor progression by regulating angiogenesis and metabolism[39,41]. While NRF2 is considered an attractive target for cancer treatment[58], our data suggest that NRF2 independent mechanisms also play a role in cancer progression. In our study, analysis of MAFF binding proteins using mass spectrometry identified a significant interaction between MAFF and BACH1 in breast cancer cells (Fig. 7). Co-immunoprecipitation further validated that MAFF binds to BACH1 and that this binding is enhanced at lower oxygen concentrations. Unlike previous reports, we do not see an increase in BACH1 levels under hypoxia, which indicates a more prominent role of MAFF at low oxygen concentrations[59]. Recent studies propose that BACH1 acts more than a competitor to NRF2 and instead may act as an activator of metastasis and metabolism[44,46,47]. Both BACH1 and MAFF depletion result in decreased metastasis in multiple preclinical models, suggesting that these proteins work together to regulate a distinct set of genes in the metastatic process.

In metabolic studies, we did not see changes in lactate levels or extracellular acidification rates as measured by the Seahorse FX analyzer, indicating that MAFF is not involved in glycolytic pathways (Supplementary Fig. 3). However, we did find that pyruvate levels were significantly decreased when MAFF was absent. These observations seemed to be consistent with the recent study showing that BACH1 negatively regulates mitochondrial metabolism[44]. Since BACH1 downregulates electron transport chain (ETC) genes, MAFF might contribute to this repression by dimerizing with BACH1.

In our study, cells with BACH1 knockdown indicate that BACH1 is responsible for tumor invasion and angiogenesis through an IL11-STAT3 pathway. Knocking down BACH1 or both MAFF and BACH1 decreased tumor cell invasion and tube formation of HUVEC cells. However, when we overexpress IL11, the decreases in cell invasion and tube formation were reversed, indicating that BACH1 is required to activate IL11 along with MAFF.

Until now, the role of MAFF in cancer has been poorly recognized. In this study we have shown that MAFF as a HIF-1-dependent transcription factor regulating tumor invasion and metastasis. Considering that NRF2 and BACH1 expression levels are unchanged under hypoxia, MAFF is an important mediator of

gene activation and repression in response to microenvironment changes in oxygen. In addition, MAFF is an indispensable binding partner of CNC and BACH proteins, and impacts multiple pathways that are necessary for tumor progression. Taken together, our findings highlight the importance of MAFF as a transcription factor regulating tumor progression under normoxia and hypoxia (Fig. 9).

## Methods

**Cell lines and culture conditions**. Cell lines were maintained at 37 °C in the humidified incubator with 5% $CO_2$. Media were used Dulbecco's Modified Eagle Medium for MDA-MB-231, MCF7, A549, PC3, HCT116, HepG2, A172, and SiHa and RPMI 1640 for OVCAR8, and SKOV3, which are supplemented with 10% FBS and 1% Penicillin–Streptomycin.

For hypoxia treatments, cells were plated at the desired density 24 h before placement in a hypoxia chamber (Invivo2-400; Ruskin Technologies) maintained at 0.5% oxygen for 24 hours.

**siRNA, shRNAs, cDNAs, and CRISPR transfection and infection**. ONTARGET *plus* SMARTpool siRNAs and shRNA constructs in the pLKO.1 vector were purchased from Horizon Discovery or MilliporeSigma. CRISPR/Cas9 All-in-One Lentivectors were acquired from applied biological materials (abm). Human MAFF and IL11 cDNAs were expressed using mammalian gene collection (MGC) in the CMV-SPORT6 vector (Horizon Discovery). To establish stable cell lines these vectors were cloned into lentiviral vector, PLX304 using attB1/attB2 sites for Gateway system.

shRNA-insensitive MAFF cDNA was also generated for rescue experiment. After MAFF cDNA was cloned into lentiviral vector, mutations were made in the third positions of codons corresponding to the region targeted by shRNA D2 (CGTGTGCCAGAAGGAGGAGCT → CGTGTGTGTCAAAAAGAGGAGCT) using Quick change II site-directed mutagenesis kit (Agilent #200523). Target sequences for shRNAs or CRISPR/Cas9 are included in Supplementary Data 3.

To generate tdTomato labeled MDA-MB-231 cells, we customized the lentiviral vector, pLV-Puro-CMV-Luciferase-T2A-dTomato expressing luciferase and tdTomato from VectorBuilder.

**esiRNA libraries**. Customized esiRNA libraries were purchased from MilliporeSigma. For invasion assay using collagen, $1.25 \times 10^3$ tdTomato labeled MDA-MB-231 cells were mixed with 25 μl of Corning® Collagen I, Rat Tail (Corning #354236) on ice and then transferred into the well of the pre-cooled 96-well plate. After polymerization, 5pmol of each esiRNA was transfected into cells. After 1week incubation, stacks of images were acquired with 10 μm Z-spacing using a fluorescence microscope (Leica DMi8). For each well, 3 images were taken with 10-power magnification. Later stacked images were converted into the binary image and analyzed using Angiogenesis Analyzer for ImageJ[60].

For cell survival assay, we used sulforhodamine B (SRB) assay[61] to minimize artifacts from metabolic changes under hypoxia.

**Western blot**. Cells were washed with PBS and protein was extracted by adding RIPA buffer supplemented with Protease inhibitor (Thermo Scientific #A32955) and phosphatase inhibitor (PhosSTOP Roche 04906845001). Protein concentration was determined using Pierce™ BCA protein Assay kit (ThermoFisher Scientific #23227). 30 μg of protein samples were boiled at 95 °C for 5 min and loaded into 4–12% Bis-Tris Plus gels (ThermoFisher Scientific #NW04120BOX or ##NW04125BOX) to run in MOPS SDS running buffer (HIF-1α, HIF-2α, HIF-1β, BACH1, NRF2, phospho-STAT3) or MES SDS running buffer (MAFF, MAFG, MAFK). After transfer using Trans-Blot Turbo system (BioRad), membranes were incubated overnight with primary antibodies. After washing three times, secondary antibodies were incubated for 1 hour. Membranes were washed three times again, and protein bands were visualized using SuperSignal™ West Dura Extended Duration Substrate (ThermoFisher Scientific #34075) or Femto Maximum Sensitivity Substrate (ThermoFisher Scientific #34094). Image Lab 4.0 was used to collect western blot images from Chemidoc.

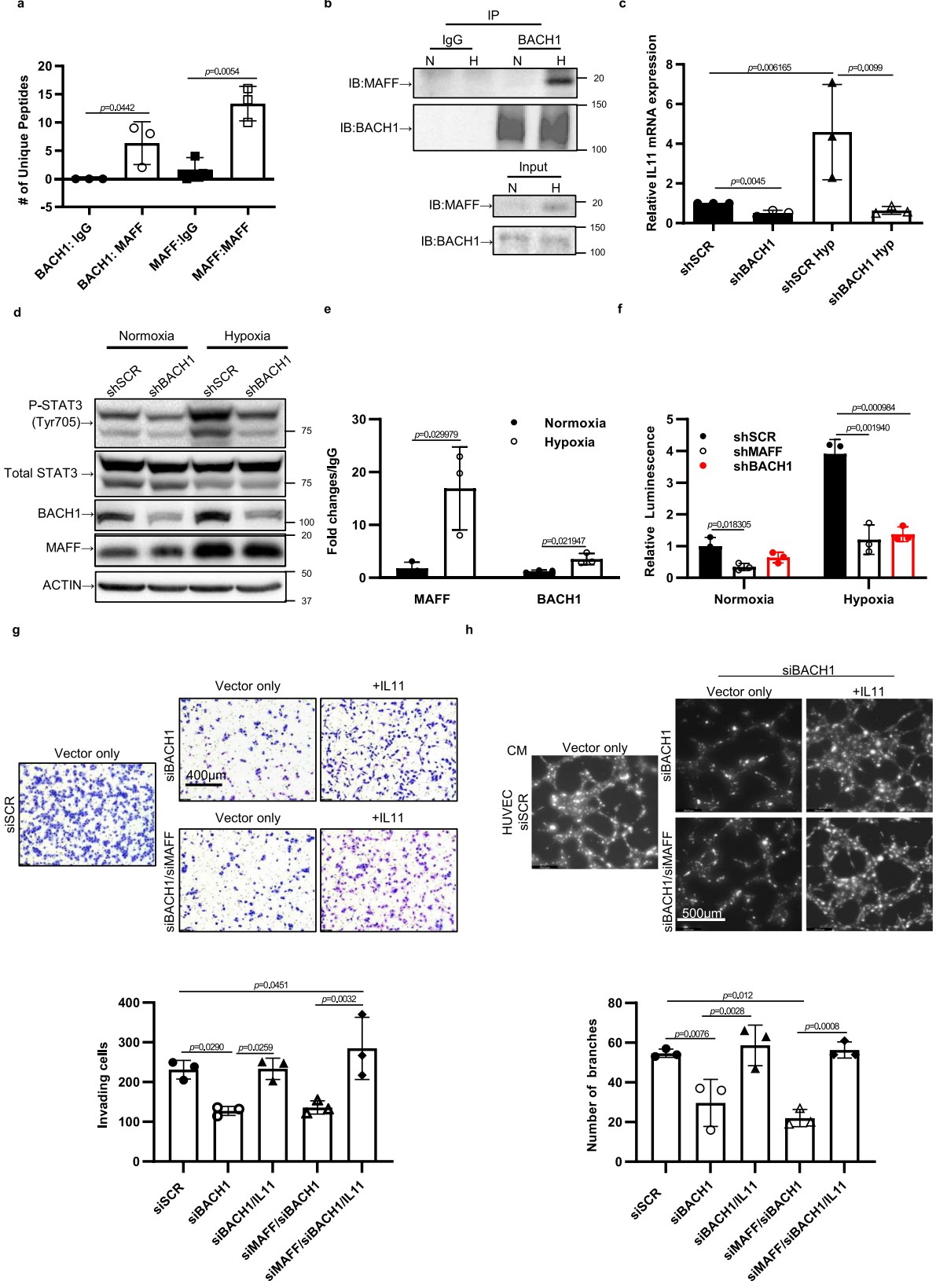

**Quantitative reverse transcription real-time PCR (qRT-PCR)**. Total mRNA was extracted using Trizol. 1 μg of mRNA was reverse transcribed into cDNA using an iScript cDNA synthesis kit (BioRad). Quantitative real-time RT-PCR was performed with iTaq™ Universal SYBR Green Supermix (BioRad) using the ABI 7900HT Fast Real-Time PCR System (Applied Biosystems). Each primer sequence is indicated in Supplementary Data 4. The expression level of each mRNA was normalized to 18s expression in the same sample.

**HIF-1α chromatin immunoprecipitation**. The ChIP assay was performed as described previously[62]. ChIP was performed on MDA-MB-231 cells exposed to normoxia (21% $O_2$) or hypoxia (0.5% $O_2$) for 24 h using rabbit anti-HIF1α antibody (Abcam). Normal rabbit IgG (Santa Cruz Biotechnology) was used as a nonspecific IgG control. Cells were formalin fixed, lysed, and sonicated using a Bioruptor Plus Sonication System (Diagenode Inc.) set at high power. Sonicated chromatin was incubated overnight with antibodies followed by precipitation with

**Fig. 7 MAFF binds to BACH1 to activate IL11-STAT3 pathways. a** MAFF binding proteins were immunoprecipitated using MAFF antibody and analyzed by mass spectrometry. In MDA-MB-231 cells, BACH1 was a main binding partner of MAFF. The graph indicates the unique peptide number identifying MAFF binding compared to IgG control. $n = 3$ biological replicates. Unpaired $t$-test. **b** Co-immunoprecipitation confirmed the binding of MAFF and BACH1, which was further increased under hypoxia. **c** When *BACH1* was knocked down, mRNA expression of *IL11* was decreased. $n = 3$ biological replicates. Unpaired $t$-test. **d** The downstream effect of BACH1 on the STAT3 pathway was determined by western blotting after *BACH1* was knocked down in MDA-MB-231 both under normoxia and hypoxia. **e** ChIP-qPCR confirmed that MAFF and BACH1 bindings on the MARE sequence of *IL11* promoter, which was identified from ChIP-sequencing. Bindings were further increased under hypoxia. Unpaired $t$-test. **f** Luciferase assay was performed using pGL4-lucirerease vector, which was subcloned with the identified MARE sequence on *IL11* promoter. While hypoxia increased the luciferase activity, *BACH1* or *MAFF* knockdown decreased the transcriptional activity of the identified MARE. $n = 3$ biological replicates. Unpaired $t$-test. Knocking down *BACH1* or both *MAFF* and *BACH1* decreased tumor cell invasion through Matrigel coated transwells (**g**) and tube formation of HUVEC representing angiogenic features (**h**) (CM: cell media). However, IL11 overexpression rescued the reduction in cell invasion and tube formation. $n = 3$ biological replicates. One-Way ANOVA with multiple comparisons. Graphs represent the mean per group and error bars represent the SEM.

protein A/G Dynabeads (Invitrogen). Five percent of the sample from each immunoprecipitation was reserved for input control. Alternatively, relative enrichment was measured by quantitative RT-PCR using a titration of pooled input samples as a standard curve and normalized to input after subtraction of IgG signal. Enrichments are presented as fold change relative to normoxia or IgG control. Each primer sequence is indicated in Supplementary Data 5.

**Transwell invasion assays**. Matrigel-coated transwell chambers were pre-incubated with serum-free media 2 h prior to experiments (Corning®). A total of $2.5 \times 10^4$ cells were placed in the upper chamber of the transwell setup and incubated for 24 h. The membrane was fixed and stained using the three step staining kit (Richard-Allan Scientific™). Invading and migrating cells were counted using bright-field microscopy.

**Collagen invasion assay**. A total of $1.25 \times 10^3$ cells were suspended in a 1:1 ratio of media and Rat Tail Type I collagen (Corning®). Cells were plated into 96-well plates. After 2 h when collagen was polymerized, 100 µl of cell growth media was added. Cells were incubated for 5–10 days to take images.

**In vitro tube formation assay**. $10^4$ cells were plated on a 96-well plate with endothelial cell medium containing 5% FBS (Cell Biologics #M1168). After 24 h incubation under normoxia or hypoxia, media was collected and briefly spun down at $200 \times g$ for 5 min to remove cell debris. 50 µl of Corning® Matrigel® Growth Factor-Reduced basement membrane matrix (Corning 354230) was added to a 96-well plate on ice. After 1 h incubation to polymerize at 37 °C, $0.5 \times 10^4$ GFP-labeled human umbilical vein endothelial cells (HUVEC) (Angio-Proteomie #cAP-0001GFP) in 100 µl of collected cell media were plated onto matrigel. 4 h after tube formation was imaged using a fluorescent microscope (Leica DMi8) with 10-power magnification.

**Luciferase reporter assay**. pGL4.42[luc2P/HRE/Hygro] vector was purchased from Promega (#E4001). While it was used as a hypoxia control, 4x HREs were excised using XhoI and BglII. Sequences of HRE8, HRE9, or mutated HRE8 or HRE9, as well as, MARE were ordered from IDT with sticky ends of overhangs of enzyme restriction. 100 µM of sense oligos and 100 µM of antisense oligos were mixed with annealing buffer and incubated for 5 min at 95 °C to anneal. Annealed oligos were diluted to 1:20 and ligated into a vector cut with two enzymes using T4 DNA ligase. The reaction was performed overnight at room temperature. Ligated DNA was transformed into DH5 alpha competent cells (ThermoFisher Scientific #18258012) and colonies were picked and grown to prep DNA using GeneJET plasmid mini prep kit (ThermoFisher Scientific #K0502). DNA with correct sequences were used for the luciferase assay.

$10^4$ cells were seeded on a 96-well plate. Then 50 ng of pGL4 firefly luciferase vector with HRE or mutations, or IL11 promoter with MARE were transfected with 5 ng of renilla luciferase vector using X-tremeGENE9™ DNA transfection reagent (MilliporeSigma #6365787001). 48 h after transfection cells were treated under normoxia or hypoxia.

1× PLB was prepared from Dual-Luciferase® Reporter assay System (ThermoFisher Scientific #E1910). After PBS washing, 20 µl of 1× PLB was added in each well and the culture plate was shaken at room temperature for 15 min. Cell lysates were mixed with 100 µl of LAR II and transferred into a white walled 96-well plate. Firefly luciferase activity was measured using the luminometer (Biotek Synergy H1). Then 100 µl of Stop & Glo reagents were mixed and renilla luciferase activity was measured using the same luminometer. Sequences are included in Supplementary Data 6.

**Extracellular acidification rate (ECAR) measurement by Seahorse XF analyzer**. $10^4$ MDA-MB-231 cells were seeded in XF96 Seahorse plate with 100 µl of DMEM containing 10% FBS and 1% antibiotics. To hydrate the cartridge, each well of the utility plate was filled with 200 µl of sterile water. Then sensor cartridge was placed

on the top of the utility plate and sensors were submerged into the water. Assembled sensor cartridge and the utility plate was incubated in a non-$CO_2$ 37 °C incubator overnight. After water was removed from the utility plate, 200 µl of pre-warmed XF calibrant (Agilent #100840-000) was added to each well. Again, sensor cartridge was assembled with the utility plate to make sensors get submerged in calibrant. After 1 h incubation in a non-$CO_2$ 37 °C incubator, the injection ports of the sensor cartridge were loaded into the machine. Cells were washed with assay media (DMEM with 25 mM glucose, 2 mM glutamine, 1 mM pyruvate, 5% FBS, pH 7.4) and filled with 160 µl of assay media. 20 µl of prepared inhibitors (25 mM Glucose, 1 µM oligomycin, 100 mM 2-deoxyglucose) were loaded into each injection port. After calibration with a calibrant plate, sensor cartridge was again assembled with the cell plate and run with the protocol of a 3 min mix time, 3 min wait, and 3 min measurement.

**Animal studies**. All animal studies were conducted with approval by the Institutional Animal Care and Use Committee (IACUC) at Stanford University. Female nude mice (Crl:NU(NCr)-*Foxn1ⁿᵘ*, 6–8 weeks) were purchased from Charles River Laboratories. $5 \times 10^5$ MDA-MB-231 cells were suspended in 25 µl of PBS and 25 µl of Matrigel (Corning 354230) and inoculated in contralateral mammary glands. Tumor growth was measured every two days using calipers. For tail vein injection, $1 \times 10^6$ MDA-MB-231 cells in 100 µl of PBS was injected into mouse tail veins. Three months after tumor cell injection, mice were sacrificed, and lung tissues were collected for H&E staining and quantitative PCR. To set up intraperitoneal tumor metastasis, $2 \times 10^6$ OVCAR8 cells in 100 µl of PBS were injected into the intra-peritoneal cavity. Twenty-eight days after, mice were sacrificed, and all visible tumor nodules were counted and weighed following our previous protocol[19].

Spontaneous lung metastasis model was performed by injecting $2 \times 10^6$ MDA-MB-231 cells in contralateral mammary glands of female NSG mice (NOD.Cg-*PrkdcˢᶜᶦᵈIl2rgᵗᵐ¹ᵂʲˡ*/Szj, 6–8 weeks, Jackson Laboratory #005557). Cells were suspended in 37.5 µl of PBS and 37.5 µl of Matrigel to make 75 µl total. Tumor growth was measured every week with calipers and when their median size reached ~1000 mm³, mice were sacrificed to collect primary tumors, lungs, and blood serum.

**Immunohistochemistry**. Before excising the tumors, the hypoxia marker pimonidazole (60 mg/kg i.p., Hypoxiprobe, Inc) was injected. Excised tumors were fixed in formalin for 48 h and stored in 70% ethanol until paraffin embedding. Paraffin-embedded tissues were sectioned and stained as previously described[15]. Briefly, slides were baked at 65 °C for 45 min and rehydrated in Xylene, 100%, 95%, 85%, 75% ethanol for 5 min each. Antigen retrieval was performed by being submerged and boiled in 10 mM citric acid, pH 6 in a microwave for 6 min. After quenching of endogenous peroxidases using 3% $H_2O_2$ in methanol, membrane was permeabilized using 0.3% Triton-X 100. After washing in TBS for 5 min, tissues were blocked with TNB buffer (0.1 M Tris-HCl pH 7, 0.15 M NaCl, 0.5% Blocking reagent) provided by the Tyramide Signal Amplification kit (TSA kit, Perkin-Elmer). Slides were blocked for 30 min and incubated with primary antibodies for overnight. MAFF (1:200, Genetex), BACH1 (10 µg/ml, R&D), IL11 (1:200, Genetex), HIF-1α (1:200, Abcam), phospho-STAT3 (1:200, Cell Signaling), MECA-32 (1:200, BD Transduction), Ki67 (1:200, Thermo Scientific), and monoclonal antibody for pimonidazole (1:50 Hypoxyprobe MAb 1) were used. On the second day, slides were washed three times in TBS with 0.5% Tween-20 and incubated with biotin-conjugated secondary antibodies (1:200, Vector or 1:500, F(ab')₂ from Hypoxyprobe, Inc.). Slides were incubated with Streptavidin-HRP (1:100), Biotin Tyramide diluted (1:50), and again streptavidin-HRP (1:100) diluted in TNB buffer. HRP signals were visualized by applying 3,3′-diaminobenzidine (DAB).

**Image analysis**. Three representative areas were imaged using a bright field microscope (Leica DMi8) with Leica Application Suite X (LAS X). For nuclear staining (Ki67 and phospho-STAT3), images were converted to 8-bit images and threshold adjusted. Then positive nuclear staining was counted and analyzed with "analyze particles, Size (pixel^2: 10-Infinity)". MECA-32 stained vessels and lung nodules from tail vein injection were manually counted. Overlapping regions

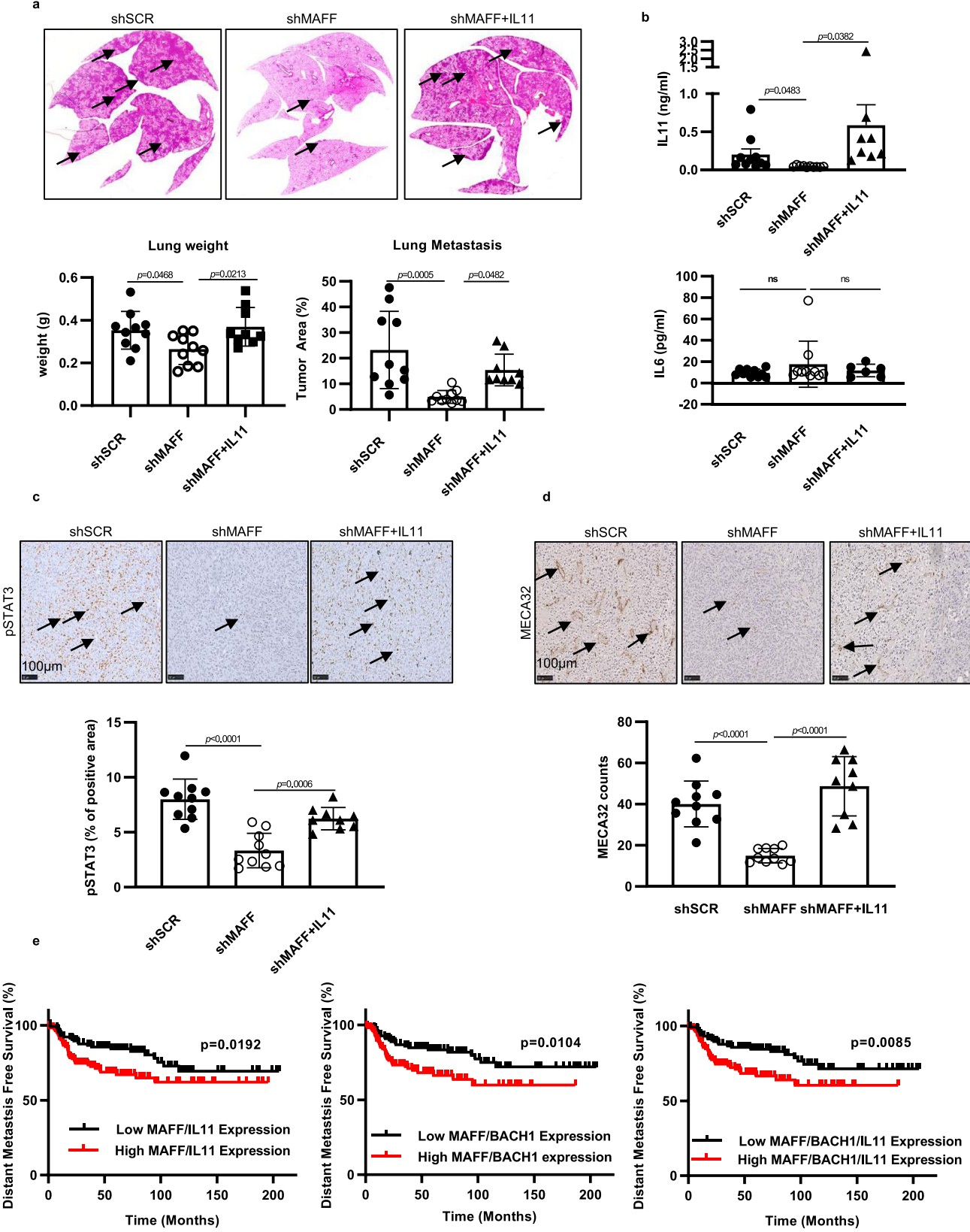

(MAFF, HIF-1, BACH1, IL11, pimonidazole) were calculated after drawing the region of interest and compare the percentage of positively stained areas, which were coexisted in both tissue sections. For spontaneously metastasized lung nodules, whole tissue sections were imaged using NanoZoomer 2.0-RS (Hamamatsu). Then using ImageJ, images were converted to 8-bit images and contrast was enhanced to saturated pixels of 0.35%. After threshold was adjusted, images were mask converted to generate to black and white images. From converted images, lung nodules were counted using "analyze particles, Size (pixel^2: 5-infinitiy)".

**Patient data**. Patient overall survival data were obtained from GEO, GSE19783 and GSE42568. Distant metastasis-free survival data were obtained from KM plotter (kmplot.com)[63]. Data were re-plotted using GraphPad.

**Fig. 8 MAFF enhances tumor metastasis through IL11/STAT3 pathways in vivo and their combined expression predicts patient survival.** Spontaneous lung metastasis model of MDA-MB-231 was studied by injecting $2 \times 10^6$ cells into the mammary fat pad of NSG mice ($2 \times 10^6$ cells/75 μl/mouse). **a** In *MAFF* knockdown group, spontaneous lung metastasis was significantly decreased while IL11 overexpression rescued the reduction. Lung weight was measured right after sacrificing mice. Also, lung tumor area was determined using ImageJ after H&E staining ($n = 10$ for shSCR and shMAFF, $n = 9$ for shMAFF+IL11). One-Way ANOVA with multiple comparisons. **b** IL11 and IL6 levels in mouse serum were determined using ELISA. While IL11 was decreased with *MAFF* knockdown, elevated IL11 levels were confirmed in shMAFF with IL11 overexpression group. However, IL6 levels remained unchanged in every group ($n = 10$ for shSCR, shMAFF, $n = 7$ or 6 for shMAFF+IL11). Unpaired *t*-test. **c, d** Expression of phospho-STAT3 and MECA32 were evaluated from primary tumor tissues and we observed consistent changes with in vitro studies suggesting that MAFF and IL11 pathways regulated phospho-STAT3 and angiogenesis ($n = 10$ for shSCR, shMAFF, $n = 9$ for shMAFF+IL11). Arrows indicate positive MECA-32 staining for vessels. One-Way ANOVA with multiple comparisons. **e** Distant metastasis-free survival of breast cancer patients was evaluated using KM plotter[63]. Combined expression of *MAFF* and *IL11*, *MAFF*, and *BACH1*, or *MAFF*, *IL11*, and *BACH1* showed the correlation with patient survival, indicating prognostic significance of these factors. Log-rank. Graphs represent the mean per group and error bars represent the SEM. ns: not significant.

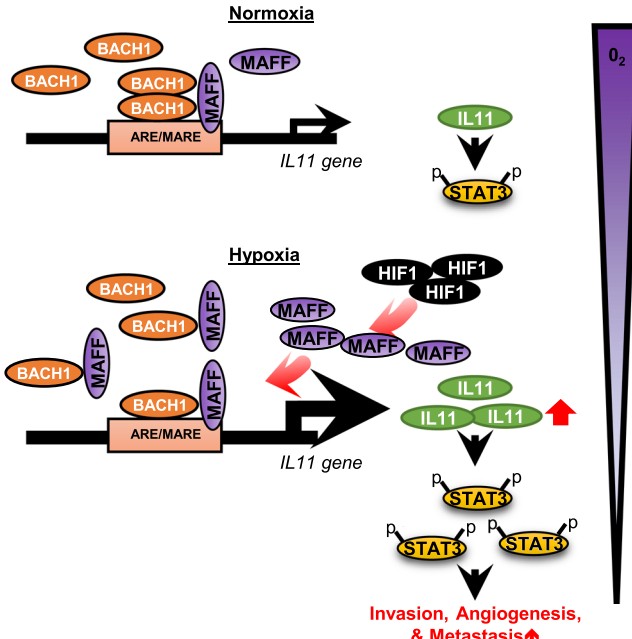

**Fig. 9 Summary of hypoxia-induced MAFF and its role in tumor invasion and metastasis through IL11/STAT3 pathways.** Hypoxia-induced MAFF binds to BACH1 to transcriptionally activate IL11. Increased IL11 leads to elevated phosphorylation of STAT3, which results in enhanced tumor invasion, angiogenesis, and metastasis.

Tissue microarray slide was purchased from Biomax (#BRM961) and stained with MAFF. MAFF staining intensity was analyzed using FIJI (Image-J-based open-source software). The color deconvolution feature was used to extract the positively stained areas. Integrated density values were calculated from the resulting binary image.

Gene expression patterns from human tumor samples were separated by their similarity to our developed hypoxia signatures[16]. Binary regression from this analysis showed highly statistically significant correlations between hypoxia signature MAFF mRNA levels.

**RNA-sequencing and ChIP-sequencing.** MDA-MB-231 cells were incubated under normoxia (21%) and hypoxia (0.5%) for 24 hours with or without siRNA against MAFF. Using QIAShredder and RNeasy Mini kit (Qiagen #79654 and #74104), RNA was extracted. For RNA-sequencing, Libraries were prepared using a TruSeq RNA sample preparation Kit (Illumina #RS-122-2001) according to the manufacturer's protocol. Samples were pooled and 50 bp reads were generated on an Illumina HiSeq2500 Analyzer (Psomagen, Maryland, USA). Raw FASTQ sequence files obtained from the RNA-seq experiments were analyzed in Base-space's RNA express pipeline (RNA Express Legacy; version: 1.0.0), using the STAR aligner and DESEQ2 to call differentially expressed genes. Genes with an adjusted *p*-value ≤0.1 were then used for further analysis. To derive MAFF-dependent genes, we included genes that were significantly regulated by MAFF siRNA in 1.3-fold changes compared to scrambled siRNA.

For ChIP-sequencing, V5-tagged MAFF was overexpressed in MDA-MB-231 cells with MAFF knockdown and chromatin was prepared as described previously[64]. $2 \times 10^7$ cell were crosslinked with 1% formaldehyde in CiA fixation buffer (50 mM HEPES pH 8, 1 mM EDTA, 0.5 mM EGTA, 100 mM NaCl) for 10 min. Then glycine was added to a final concentration of 125 mM to stop the fixation. To collect nuclei cells were resuspended in CiA NP-Rinse 1 (50 mM HEPES pH 8, 140 mM NaCl, 1 mM EDTA, 10% glycerol, 0.5% NP40, 0.25% Triton X-10) and incubated in ice for 10 min. Cells were washed with CiA NP-Rinse 2 (10 mM Tris pH8, 1 mM EDTA, 0.5 mM EDTA, 200 mM NaCl) and dissolved in Covaris shearing buffer (0.1% SDS, 1 mM EDTA, 10 mM Tris-HCl pH 8). Cells were sonicated using Covaris S220 (Intensity 4, Duty Factor 5%, Cycle/Burst 200) for 420 s. After spinning down, supernatants were collected for immunoprecipitation using V5 antibody (Cell Signaling #13202) and Dynabeads protein A (Thermo Fisher Scientific #10001D) overnight at 4 °C with rotation. Using magnetic stand, beads were collected and washed with IP buffer (50 mM HEPES.KOH, 300 mM NaCl, 1 mM EDTA < 1% Triton-X, 0.1% DOC, 0.1% SDS), DOC buffer (10 mM Tris pH8, 0.25 M LiCl, 0.5% NP40, 0.5% DOC, 1 mM EDTA), and TE pH 8. Beads were mixed with TE, 0.5% SDS, and proteinase K and incubated overnight at 65 °C to reverse crosslink. ChIPed DNA was collected from beads and eluted using QIAquick PCR purification kit (Qiagen #28104).

Libraries were prepared using NEBNext Ultra II DNA Library Prep kit for Illumina (NEB #E7103) and NEBNext Multiplex Oligos for Illumina (NEB #E7730S). To control the quality of DNA libraries, the size and purity of samples were confirmed by Bioanalyzer. Samples were pooled and 2x150bp paired-end sequencing was performed by Illumina HiSeq2500 Analyzer (Genewiz, New Jersey, USA). Raw FASTQ files obtained in the ChIP-seq experiments were aligned with Bowtie2 and MACS to call peaks. The cutoff value used was *q*-value < = 0.05. MEME was used for de novo motif analysis of the genomic DNA sequences flanking ChIP-seq peaks. The percentage of peaks containing a MAFF consensus motif was determined with RSA-tools matrix scan.

GSEA, DAVID Bioinformatics Resources 6.7[20,21], and Pubmatrix were used to analyze the gene lists derived from RNA-seq or ChIP-seq data to find enriched pathways and biological functions[21]. All RNA-seq and ChIP-seq data are available in the GEO database (http://www.ncbi.nlm.niv.gov/gds) under the accession number GSE144964.

**Co-immunoprecipitation.** Nuclear protein was extracted using a modified protocol from the previous publication[65]. After washing in cold PBS, hypotonic buffer (10 mM HEPES, pH 7.9, 10 mM KCl, 1.5 mM $MgCl_2$, Roche cOmplete™ protease inhibitor cocktail (MilliporeSigma 11697498001)) was added to cells. Cells were scraped and homogenized 10 times with a dounce homogenizer (pestle B) (Kimble Chase). Samples were spun down at $17,000 \times g$ at 4 °C for 5 min. After brief wash with hypotonic buffer, pellets were resuspended in high salt buffer (20 mM HEPES, pH 7.9, 0.42 M NaCl, 25% glycerol, 5 mM $Ca_2Cl$, 1.5 mM $MgCl_2$, 0.1% Nonidet P-40 (NP-40), 0.2 mM EDTA, Roche cOmplete™ protease inhibitor cocktail). Samples were homogenized 20 times with a dounce homogenizer (pestle B). Chromatin was further digested by incubation with micrococcal nuclease (ThermoFisher Scientific #88216) at room temperature for 10 min. Samples were centrifuged at $17,000 \times g$ at 4 °C for 15 min and supernatants were collected.

Samples were incubated with antibodies (MAFF: 15 μg, MilliporeSigma #M8194, BACH1: 5 μg, Santa Cruz #sc-14700, or IgG: Santa Cruz #sc-2027) overnight at 4 °C and using Pierce protein magnetic beads A/G (Thermo Scientific), protein was pulled and eluted.

**Mass spectrometry.** To determine protein-protein interaction, eluted samples were run on 4–20% precast protein gel (BioRad #4561094). When samples were run about 1 cm, the run was stopped and gel was stained by Coomassie Blue. After washing, gel was cut into 1 cm × 1 cm cubes and samples were processed and analyzed at Stanford Mass Spectrometry Facility following the previously published protocol[66].

**Statistical analysis and reproducibility.** All experiments were performed for at least two times. Student's *t*-test, log-rank test, one-way, or two-way ANOVA with

post-hoc Fisher's PLSD tests were performed to determine statistical significance using GraphPad. All error bars represent the mean ± SEM.

**Reporting summary**. Further information on research design is available in the Nature Research Reporting Summary linked to this article.

## Data availability

The RNA and ChIP-sequencing data (Fig. 5, Supplementary Fig. 5) have been deposited in the GEO database under the accession number GSE144964. The source data underlying western blotting are provided as a Source Data file. All the other data supporting the findings of this study are available within the article and its Supplementary information files and from the corresponding author upon reasonable request. cbioportal was used to determine genetic alterations in hypoxia-regulated genes (cbioportal.org). Kaplan–Meier plotter (kmplot.com), and GEO datasets: GSE19783, GSE42568 were used to evaluate patient survival. Source data are provided with this paper.

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

## Acknowledgements

We acknowledge Ryan Leib, PhD and Christopher M. Adams, PhD from Vincent Coates Foundation Mass Spectrometry Laboratory, Stanford University Mass Spectrometry. This work was supported by NIH P01 CA67166 and in part by NIH P30 CA124435 utilizing the Stanford Cancer Institute Proteomics/Mass Spectrometry Shared Resource.

## Author contributions

Conceptualization, E.J.M. and A.J.G.; methodology, E.J.M. and A.J.G.; investigation, E.J.M., S.S.M., C.G.L., J.C., K.T., J.K., E.L.L., A.N.D., M.R., I.J.L., M.V., Y.M., L.C., and A.J.K.; writing-original draft, E.J.M.; writing—review & editing, A.N.D., M.R., and A.J.G.; supervision, E.E.G., L.D.A., and A.J.G.; funding acquisition, A.J.G.

## Competing interests

The authors declare no competing interests.
