## [Peer Review File · Nature Communications]

Reviewers' comments:

Reviewer #1 (Remarks to the Author):

This manuscript by Moon at all investigates a potential mechanism of breast cancer metastasis via MAFF/BACH1 interaction as well as subsequent production of IL11 and STAT activation. In general the in vitro data presented are of high quality with appropriate controls. The main gaps in the manuscript are related to 1. In vivo validation of the proposed mechanism and 2. Incomplete assessment of the mechanism itself. These areas are discussed in detail below.

In vivo validation

The authors use tail vein injection and IP injection of tumor cells into nude mouse hosts to model metastatic disease. These assays do not recapitulate the metastatic cascade. They can simulate colonization of distant tissues. If the authors wish to use these assays in the absence of an additional model that undergoes full metastasis they should be careful to amend the text so that it is clear to readers that one cannot draw conclusions about metastatic progression from a primary tumor using tail vein injections of tumor cells directly into vasculature. Further confounding this issue, the authors use lymph node metastases in a tissue microarray (Fig. 1E) to demonstrate high expression of MAFF in metastatic nodes. This is appropriate given that breast cancers are known to spread through the lymphatics. However, that finding simply highlights the issues associated with tail vein injection approaches, which do not involve the lymphatic system only the endothelial vasculature. Moreover, all the invasion assays and microvessel density experiments documented in the manuscript study how MAFF and IL11 relate specifically to tumor and endothelial cell function in the primary microenvironment. It is impossible to know whether in vivo observations made using tail vein and IP injections recapitulate mechanisms associated with those aspects of primary tumor progression. If 231 and MCF-7 cells do not metastasize from orthotopic injection sites then perhaps organotypic culture or patient-derived xenografts will provide the necessary systems to observe hypoxia-mediated invasion and vascular metastasis.

Incomplete assessment of the mechanism

The authors present compelling data linking MAFF and BACH1 to IL11 expression and STAT3 activation. However, the authors have only performed rescue assays using MAFF depleted cells reconstituted with IL11 to rescue invasion, STAT3 phosphorylation and microvessel density. They have not performed similar assays to rescue BACH1 loss with IL11. Does IL11 does not rescue these phenotypes in BACH1 deficient cells? what about rescue of BACH1/MAFF double knockdown with

IL11 expression? These experiments will allow interpretation of the true contribution of IL11 signaling downstream of BACH1/MAFF. Moreover, the authors do not show the role of STAT3 in invasion/ microvessel density or in vivo metastasis. Without these links the branches of the proposed mechanism appear disconnected from one another.

Reviewer #2 (Remarks to the Author):

Hypoxic response is known to promote metastasis of cancer cells. In this manuscript, authors identified one of the small MAF oncogenes, MAFF, as a target of hypoxia transcription factor HIF1. Using various human cancer cells lines in vitro and mouse xenograft models, authors showed that MAFF is required for invasion activities and metastasis. Authors then showed that MAFF promotes not only metastasis but also angiogenesis by inducing the expression of IL11. Finally, they showed that MAFF formed a heterodimer with BACH1, not with NRF2, to promote these responses. This manuscript is well organized, comprehensive and with new findings. However, several issues need to be addressed to validate their conclusions.

Major

1. Authors used antibody against MAFF. However, it is very similar to MAFK and MAFG in amino acid sequence. Thus, validation of the antibody specificity is essential as a starting point. For this, they can knockdown each of the genes to see if the signal of immunoblotting is affected or not. Also, mRNAs will be quantified to exclude possible cross-knockdown. (siRNAs need to be targeted to the 3' untranslated regions of these mRNAs.)
2. To identify MAFF target genes, authors combined the data of RNA seq and chromatin IP. Please describe how many genes were identified as both 'expression altered upon siRNA of MAFF' and 'bound by MAFF'. This is essential to grasp overview of putative direct target genes.
3. For IL11 and other few genes, distribution of tags along genes (genome viewers) must be depicted as a main figure (both normoxia and hypoxia). For other genes, show genome viewer images as a supplementary figure.
4. In addition, binding of MAFF and BACH1 to peak regions must be validated by quantitative PCR after CHIP. Also, a reporter assay should be established to see if co-expression of MAFF and BACH1 activate or not. This will be simple but very important.
5. MDA-MB-231 transplantation into fat pad is known to cause metastasis. What happened upon knockdown of MAFF? This information is important along the data in Fig. 4c.

6. Are HRE8 and 9 really responsive to HIF1 to activate MAFF? A simple reporter assay would answer this question.

Others

7. on line 53, the original paper on the discovery of small MAF should be cited (Oncogene 1993, 8(9):2371-80.).

8. on line 63, authors did not cite papers on BACH1-MAF interactions. Some of the references cited later may be cited here as well.

9. Fig 1b is strange. Is this describing expression of the 50 genes or one of them as an example? If the expression of 50 genes were aggregated, how was it carried out? Text and legend can be improved.

10. What is esiRNA?

11. Fig. 1e, do authors mention about the frequencies of positive cells or staining intensities per cell?

12. Fig. 1f is overinterpreted. There is no correlation value indicated or its significance. Avoid 'MAFF is strongly linked to hypoxia' in the text. How was the hypoxic feature defined? Rather, authors may need to carry out multivariate analysis including clinical parameters.

13. on line 176, they are not target genes at this stage.

14. Positions of molecular weight markers should be given especially for MAFF and BACH1 in the immunoblotting images.

15. Figure 3c and f, upper and lower panels should be explained in the figure legends.

16. On line 96 "OS, $p=0.0194149$; MFS, $p=0.007587$ " are not consistent with the values in the figure.

17. On lines 140 and 206, what other cells the expression levels of Nrf2 were compared with?

18. On line 171, how long was the hypoxia treatment? 16 hrs in the text but 24 hrs in the figures.

19. On line 244, cite figure 7b.

20. On lines 250-255, figure panels are wrongly cited.

21. On line 315, mediator not mediator.

22. On line 459, provide programs for testing. In the figure legends, describe which tests were used.

23. Fig. 1c, GO rather than GEO.

24. Fig. 2 and other figures, use unified names for genes or proteins.

25. Fig. S2c, values of expression amounts in these cells are lost by dividing hypoxia values with normoxia values.

26. Fig. 3e,e,f, brief description of culture conditions in the legend will help understanding.

27. Fig. 4d, how were tumor weights determined? Also, representative pictures will help.
28. Fig. 5, ChIP-QPCR to examine increased binding of MAFF to some of the target genes should be shown.
29. No legend for Fig. 6e.
30. Fig. 7f, Do IL11 high and low groups show any difference? This is important to establish the suggested role of IL11 in tumor metastasis and prognosis of patients.
31. Fig. s1, no description of statistical test in the legend.
32. Fig. s6b, did MAFF knockdown remain effective after IL11 transfection? RT-PCR?

Kazuhiko Igarashi

Reviewer #3 (Remarks to the Author):

The authors of the study entitled “The HIF target MAFF promotes tumor invasion and metastasis through IL11 and STAT3 signaling” start their work from a RCC4 clear-cell renal cell carcinoma line screen on hypoxia-relevant genes potentially changing metastasis. They evaluate publically available data to investigate the most significant expression changes of candidate genes in various solid cancers. Subsequently, due to negative prognostic values they focus on the transcription factor MAFF in breast cancer (BCa) that lacks a transactivation domain, but functions in association with coactivator or corepressor, where they identify in a nice MS approach following affinity chromatography interaction with BACH. The authors conclude that a BACH1-MAFF interaction is crucial for invasive properties of BCa cell lines and this cell line in vitro work insight is solid. Less solid is the lack of proliferation change and a too limited in vitro cell line analysis where the reviewer suggest more insightful doable experimentation. Overall, the study has mechanistic power and it is relevant with illuminating the clinical dilemma of metastasis. A drawback comes from experimental systems that rely heavy on cancer cell lines and their genetic manipulation, sometimes experimental setup to reveal proof for metastasis is not really state-of-the-art, questioning relevance. Here, more should be done for a Nature Communication article to substitute the major conclusion that the BACH1-MAFF interaction is crucial for metastasis. To extend on in vitro analysis a simple analysis on lactate and pyruvate levels in cancer cell lines should be undertaken and the metabolic controls of these proteins could be better placed in context for writing the discussion. Last but not least, IL-11 autocrine loops might be a too narrow view and IL-6 trans-signaling could very well be relevant. In summary, the study could benefit from doable experiments, plus an improved discussion before publication.

Major:

1) Cell line work allows for cancer cell metabolism insights and that could influence e.g. through mTOR signalling and S6 kinase steering cell size control. STAT3/5 transcription factors can induce expression of cytokines, growth factors, chemokines or their cognate receptor chains, thereby regulating innate and acquired immunity or to promote tumor surveillance. The authors focus here on hypoxia gene expression also changing tumor vasculature and in this regard it is important to discuss that HIF1 α / β /VEGF family members can be under control of STAT3/5 transcription factors in breast cancer cells. Furthermore, metabolic control by serine phosphorylated STAT3 downstream of RAS transformation or oncogenic RAS signaling in general will be switched on in most if not all of these used carcinoma cell lines. This will probably promote through mitochondrial function of serine phosphorylated pSSTAT3 or through transcriptional activation of tyrosine phosphorylated pYSTAT3 it (especially if mitochondria are damaged and not functional). One would expect significant changes in levels of pyruvate and lactate production leading to promotion of angiogenesis dependent on normoxic or hypoxic conditions, associated with the Warburg effect, change of lactate transporter genes, etc. promoting at the end the surviving of cancer cells under hypoxic conditions and promoting tumor neovasculature. The authors should at least measure upon genetic manipulation with high or low MAFF expression levels of pyruvate and lactate production in their tissue culture systems. Normally, proliferation is associated with high metabolism rates and arrest of proliferation was reported to happen for metastasis, the authors report no change in growth of their cancer cell line systems, but the reviewer is not so convinced here that a metabolic reprogramming does not happen here. Metastatic cells could lose contact to the plastic plate, subsequently go into solution and die quickly off, authors might miss it which could lead to misinterpretation of results. Here, paired model BCa cell lines under normoxic or hypoxic conditions are suitable to get an insight on simple lactate and pyruvate levels that are under control of STAT3 and should be changed to the understanding of the reviewer under hypoxic or normoxic conditions. Involvement of metabolic change by seahorse measurement is also a possible alternative readout. Possible change of metabolism could be better placed in experimental context as well as discussed.

2) The authors perform not really a metastasis assay that is physiologic and relevant. A changed larger cell size would influence (particularly in an artificial setup like here with tail vein injection) invasion or clogging of BCa cells in lung vessels. In contrast smaller cell size would lead to better clearance. A better metastasis read out for the crucial MAFF genetic cell line pair could be investigated, changed mechanical properties of injected controlled cell size cell line pairs need to be shown. At the end blood vessels in the lung are the smallest and e.g. if cancer cells have different size (in association with suspected change in metabolism), which could e.g. be controlled by simple forward side scatter analysis in a FACS machine that used cell lines are indeed similar of size profiles then the largest carcinoma cells get blocked in the smallest vessels by mechanical principles. That is not really a physiologic BCa metastasis assay, the authors should look up e.g. a recent Nature paper of STAT3 role in liver metastasis for a GEMM(<https://doi.org/10.1038/s41586-019-1004-y>) in systems of carcinogenesis how animal model set up can be investigated.

In BCa research a quick read out animal model system are e.g. Rag2 $^{-/-}$ γ c $^{-/-}$ or NSG mice on a Balb/C background, which are susceptible for BCa metastasis assays in short read out times. One could e.g.

take human BCa cell lines such as e.g. 1x10⁶ MDA-MB-231 could be injected into the 4th mammary fat pad of female immunosuppressed mice. Here authors should make use of their paired MAFF genetic cell system with stable high or low MAFF expression. Alternatively, also an immunocompetent animal model that relies on transplant can be used e.g. with 1x10⁴ 4T1 cells (where authors should genetically manipulate MAFF expression) injected into the 4th mammary fat pad of female wildtype Balb/C mice. When tumors reach about 100 mm³ in volume, the primary tumors need to be resected. Ten days after primary tumor removal, mice need to be sacrificed, lungs need to be isolated and the lung weight should be assessed as well as lung pathology using specific epithelial and EMT cell markers would be most insightful, where e.g. VIMENTIN, CK8, KI-67, CLEAVED CASPASE-3 Hematoxylin and Eosin G should be carried out for minimal histo-pathology analysis to provide a more physiologic BCa test system.

Minor:

3) A model could be drawn in the last part of the study where the reviewers suggests to draw an oncogenic transcription factor network such as MAFF-BACH interaction, STAT3 and HIF1 α/β could be interesting to readers to present in a scheme, e.g. summarizing findings in a cartoon at the last Figure part.

4) With regard to the interpretation of the xenograft experiments presented in this study it is important to note that IL-6 trans-signaling as well as IL-11 activity are both active in mice or patient derived cells giving rise to xenograft tumors in immunosuppressed mice allowing for corresponding cytokine-receptor interaction [Jostock, T., Mullberg, J., Ozbek, S., Atreya, R., Blinn, G., Voltz, N., et al., (2001) Soluble gp130 is the natural inhibitor of soluble interleukin-6 receptor transsignaling responses. *Eur J Biochem.* 268(1): p. 160-7.]. Therefore, most likely activators of STAT3 signaling in mouse xenografts are not restricted to IL-11 only and will originate also from IL-6 trans-signaling. The authors should therefore at least discuss plasma soluble IL-6R α levels and take that into account. All other gp130 cytokine ligands are species restricted. The authors have autocrine cytokine action evidence of IL-11 secretion and signaling in BCa cells, but IL-6 trans-signaling could enhance the vicious cycle, subsequently driving HIF-1 α and MAFF expression.

5) The sentence on page 7 “Overlapping regions of MAFF and pimonidazole,....” is misleading and could be reworded/improved for understanding as follows: “Overlapping regions of MAFF and hypoxic areas as labelled by the chemical pimonidazole as a hypoxic tumor area marker were frequently observed as quantified by histology scores (~87%)....”

Reviewer #4 (Remarks to the Author):

Comment:

Hypoxia plays a critical role in tumor progression including invasion and metastasis. The manuscript titled “The HIF target MAFF promotes tumor invasion and metastasis through IL11 and STAT3 signaling” by Eui Jung Moon et al. screened fifty hypoxia inducible genes, identified MAFF as a potent regulator of tumor invasion without affecting cell viability, which is elevated in metastatic breast cancer patients and is specifically correlated with hypoxic tumors. The authors further identified IL11 as a direct transcriptional target of the heterodimer between MAFF and BACH1, leading to the activation of STAT3 signaling, demonstrating the oncogenic role of MAFF as an activator of the IL11/STAT3 pathways in breast cancer. The manuscript is very well written and is well balanced. The story is interesting, and the findings are significant. However, several concerns need to be answered before considering acceptance.

Major points:

1. It lacks in vivo data to support the importance of IL11-STAT3 pathway in MAFF induced tumor cell invasion and metastasis.
2. The effect of MAFF in blood vessel development should be measured in tumor tissues. The result of knockout of IL11 is not strong enough to support MAFF's role in blood vessel development.
3. Since inhibition of MAFF suppresses tumor invasion without affecting cell viability, and the author concluded that the combined regulation of IL11-STAT3 signaling pathway by MAFF and BACH1 is responsible for MAFF induced tumor cell invasion and metastasis. However, there is not enough proof in this manuscript to show the inhibition of BACH1, or IL11-STAT3 could suppress tumor invasion without affecting cell viability.

Minor points:

1. Please describe how the survival analysis was performed in Fig 1b in detail.
2. Please explain how the intensity was measured in Fig 1e IHC data. Expression data of MAFF in primary tumor and lymph node from online dataset may be added.
3. The analysis in Fig 1f lacks p value and R value.
4. In Fig 4b, please replace “MVD” with “MECA-32”.
5. For Fig 5c, it is not clear the analysis was performed with 240 genes significantly altered under hypoxia or 160 genes regulated in both normoxia and hypoxia. Please clearly clarify it.
6. The table in Fig 7c lacks p value.

7. Correlation analysis of HIF-MAFF, MAFF, BACH1/IL11 is needed with expression data in tumor tissues.

We have addressed Reviewers' comments below.

Reviewers' comments:

Reviewer #1 (Remarks to the Author):

This manuscript by Moon at all investigates a potential mechanism of breast cancer metastasis via MAFF/BACH1 interaction as well as subsequent production of IL11 and STAT activation. In general the *in vitro* data presented are of high quality with appropriate controls. The main gaps in the manuscript are related to 1. *In vivo* validation of the proposed mechanism and 2. Incomplete assessment of the mechanism itself. These areas are discussed in detail below.

In vivo validation

The authors use tail vein injection and IP injection of tumor cells into nude mouse hosts to model metastatic disease. These assays do not recapitulate the metastatic cascade. They can simulate colonization of distant tissues. If the authors wish to use these assays in the absence of an additional model that undergoes full metastasis they should be careful to amend the text so that it is clear to readers that one cannot draw conclusions about metastatic progression from a primary tumor using tail vein injections of tumor cells directly into vasculature. Further confounding this issue, the authors use lymph node metastases in a tissue microarray (Fig. 1E) to demonstrate high expression of MAFF in metastatic nodes. This is appropriate given that breast cancers are known to spread through the lymphatics. However, that finding simply highlights the issues associated with tail vein injection approaches, which do not involve the lymphatic system only the endothelial vasculature.

Moreover, all the invasion assays and microvessel density experiments documented in the manuscript study how MAFF and IL11 relate specifically to tumor and endothelial cell function in the primary microenvironment. It is impossible to know whether *in vivo* observations made using tail vein and IP injections recapitulate mechanisms associated with those aspects of primary tumor progression. If 231 and MCF-7 cells do not metastasize from orthotopic injection sites then perhaps organotypic culture or patient-derived xenografts will provide the necessary systems to observe hypoxia-mediated invasion and vascular metastasis.

: To demonstrate the role of MAFF in a more physiologically relevant *in vivo* model, we injected MDA-MB-231 cells into the mammary fat pad of NSG mice, which form spontaneous lung metastasis within 2 months¹. Consistent with our previous observations from tail vein injection models, we found that MAFF knockdown significantly prevented tumor cells from metastasizing to the lung in this model. To further strengthen the role of MAFF promoting metastasis through an IL11/STAT3 pathways, we included an experimental group with MAFF

knockdown and IL11 overexpression. We found that overexpression of IL11 rescued the decreased tumor lung metastasis resulting from MAFF knockdown. We also included phospho-specific STAT3 staining as well as MECA32 staining to show that MAFF regulates tumor angiogenesis through IL11 and STAT3 pathways. Data are now included in Figure 8.

Incomplete assessment of the mechanism

The authors present compelling data linking MAFF and BACH1 to IL11 expression and STAT3 activation. However, the authors have only performed rescue assays using MAFF depleted cells reconstituted with IL11 to rescue invasion, STAT3 phosphorylation and microvessel density. They have not performed similar assays to rescue BACH1 loss with IL11. Does IL11 does not rescue these phenotypes in BACH1 deficient cells? what about rescue of BACH1/MAFF double knockdown with IL11 expression? These experiments will allow interpretation of the true contribution of IL11 signaling downstream of BACH1/MAFF. **Moreover, the authors do not show the role of STAT3 in invasion/ microvessel density or in vivo metastasis.** Without these links the branches of the proposed mechanism appear disconnected from one another.

: **To further validate whether BACH1 plays a role in tumor invasion, angiogenesis, and STAT3 phosphorylation, we performed rescue experiments by overexpressing IL11 in BACH1 or MAFF/BACH1 knockdown cells. In the absence of BACH1 or MAFF/BACH1, we observed decreased tumor cell invasion, STAT3 phosphorylation, and endothelial tube formation of HUVEC cells. However, when IL11 was overexpressed in these cells, all three endpoints were rescued. As we discussed above, in vivo experiments were also performed by overexpressing IL11 in MAFF knockdown cells. Using a more relevant metastasis model, we confirmed that decreased spontaneous lung metastasis with MAFF knockdown was reversed when IL11 was overexpressed. While the role of STAT3 in invasion and angiogenesis is well reported in multiple studies, we have performed experiments to specifically show the role in our experimental settings. We observed that knocking down STAT3 inhibited tumor invasion and tube formation of HUVECs. We further demonstrated that expression of phospho-specific STAT3 was decreased by MAFF, while overexpression of IL11 increased phospho-STAT3 expression in tumors as detected by immunohistochemistry. Data are included in Figure 7g and h.**

Reviewer #2 (Remarks to the Author):

Hypoxic response is known to promote metastasis of cancer cells. In this manuscript, authors identified one of the small MAF oncogenes, MAFF, as a target of hypoxia transcription factor HIF1. Using various human cancer cells lines in vitro and mouse xenograft models, authors showed that MAFF is required for invasion activities and metastasis. Authors then showed that MAFF promotes

not only metastasis but also angiogenesis by inducing the expression of IL11. Finally, they showed that MAFF formed a heterodimer with BACH1, not with NRF2, to promote these responses. This manuscript is well organized, comprehensive and with new findings. However, several issues need to be addressed to validate their conclusions.

Major

1. Authors used antibody against MAFF. However, it is very similar to MAFK and MAFG in amino acid sequence. Thus, validation of the antibody specificity is essential as a starting point. For this, they can knockdown each of the genes to see if the signal of immunoblotting is affected or not. Also, mRNAs will be quantified to exclude possible cross-knockdown. (siRNAs need to be targeted to the 3' untranslated regions of these mRNAs.)

: We have performed experiments to validate the specificity of MAFF antibody. MAFF and other small MAF proteins, MAFG and MAFK were knocked down using small interfering RNAs (siRNAs) and their expression was evaluated by qPCR and western blotting. We found that the MAFF antibody (M8194, Sigma), MAFG antibody (GTX114541, Genetex), and MAFK antibody (MAB3809, R&D) were specific for each protein and did not cross react with each other. The qPCR results further demonstrated that the siRNAs were specific for each of the MAFF genes. Data are included in Supplementary Figure 2a and b.

2. To identify MAFF target genes, authors combined the data of RNA seq and chromatin IP. Please describe how many genes were identified as both 'expression altered upon siRNA of MAFF' and 'bound by MAFF'. This is essential to grasp overview of putative direct target genes.

: We have modified the text to read "By combining profiles from RNA-sequencing and ChIP-sequencing, we highlighted 45 genes under normoxia, 44 genes under hypoxia, and 8 genes under both conditions that were directly bound and regulated by MAFF." This statement is included on page 9.

3. For IL11 and other few genes, distribution of tags along genes (genome viewers) must be depicted as a main figure (both normoxia and hypoxia). For other genes, show genome viewer images as a supplementary figure.

: We included ChIP-seq profiles of IL11 and BMP4 both under normoxia and hypoxia using the UCSC genome browser in Figure 5f. Information on the rest of genes is also included in Supplementary Figure 7.

4. In addition, binding of MAFF and BACH1 to peak regions must be validated by

quantitative PCR after CHIP. Also, a reporter assay should be established to see if co-expression of MAFF and BACH1 activate or not. This will be simple but very important.

: To validate ChIP-seq results as well as the contribution of BACH1, we performed ChIP PCR and reporter assays. The region of MAFF binding to IL11 promoter was identified from ChIP-seq and we evaluated this region for MAFF and BACH1 binding using both ChIP and reporter assays. Data are included in Figure 7e and f.

5. MDA-MB-231 transplantation into fat pad is known to cause metastasis. What happened upon knockdown of MAFF? This information is important along the data in Fig. 4c.

: We performed *in vivo* experiments on a spontaneous metastasis model that metastasizes from the mammary fat pad to the lung. Consistent with tail vein injection models, knocking down MAFF decreased lung metastasis. Data are included in Figure 8.

6. Are HRE8 and 9 really responsive to HIF1 to activate MAFF? A simple reporter assay would answer this question.

: We performed reporter assays using HRE8 and HRE9 sequences of MAFF where HIF1 binds. Consistent with the ChIP assay, we found that luciferase activity was increased under hypoxia with HRE8 or HRE9 sequences. Also, mutations on these sequences inhibited the hypoxia-induced activity of luciferase, indicating that hypoxia regulates MAFF specifically through HRE8 and HRE9 sequences. Data are included in Figure 2f.

Others

7. on line 53, the original paper on the discovery of small MAF should be cited (Oncogene 1993, 8(9):2371-80.).

: We included the reference on page 3.

8. on line 63, authors did not cite papers on BACH1-MAF interactions. Some of the references cited later may be cited here as well.

: We included the paper by Oyake et al (*Mol Cell Biol.* 1996) as a reference on page 3.

9. Fig 1b is strange. Is this describing expression of the 50 genes or one of them as an example? If the expression of 50 genes were aggregated, how was it carried out? Text and legend can be improved.

: We apologize that it was not clear how we created Figure 1b. Using the ProgGeneV2 prognostic database², we determined the prognostic significance of top 50 hypoxia regulated genes. This database automatically generates the patient survival curve using the combined

expression of the 50 genes. We have included the sentence, "Using the prognostic database, ProgGeneV2², we found that the combined expression of these genes correlated with poor patient survival, suggesting their oncogenic role for them in breast cancer (GSE19783) (Fig. 1b)" on page 4.

10. What is esiRNA?

: We have now added more details about esiRNA. The term esiRNA stands for Endoribonuclease-prepared siRNAs. Unlike chemically synthesized siRNA, esiRNA is a pool of siRNAs generated by the cleavage of long double-stranded RNA (dsRNA) by endoribonuclease. Therefore, esiRNA targets mRNA with high specificity and lower off target effects. Now we include the sentence, "To more efficiently and specifically knockdown these 50 genes, we used endoribonuclease prepared siRNA (esiRNA) libraries (MilliporeSigma), and examined the impact of each gene on invasion under normoxia or hypoxia in MDA-MB-231 cells, a human breast cancer cell line with high metastatic potential." on page 4.

11. Fig. 1e, do authors mention about the frequencies of positive cells or staining intensities per cell?

: We added more details on how we quantified MAFF staining from tissue microarrays. We measured "staining intensity" using ImageJ. After MAFF was stained and imaged, we performed color deconvolution to isolate DAB positive staining using ImageJ. Then we quantified staining intensities. Details are now included in the Methods section.

12. Fig. 1f is overinterpreted. There is no correlation value indicated or its significance. Avoid 'MAFF is strongly linked to hypoxia' in the text. How was the hypoxic feature defined? Rather, authors may need to carry out multivariate analysis including clinical parameters.

: We now included a correlation value and a p value for this data ($R^2=0.314$ and $p<0.0001$). We modified the sentence to read, "We also found that there was a statistically significant positive correlation between hypoxia signatures and MAFF levels in breast cancer patients, supporting the hypoxic regulation of MAFF (Fig. 1f)" on page 4. Data was generated using hypoxia features included in the paper published by Gatzka et al. (2011, Breast Cancer Research). Details are now included in the Methods section. Since we do not claim hypoxia as a predictor of patient survival, multivariate analysis including clinical parameters is not included.

13. on line 176, they are not target genes at this stage.

: We agree with the comment and changed the term from "MAFF target genes" to "genes"

identified from RNA seq..." on page 8.

14. Positions of molecular weight markers should be given especially for MAFF and BACH1 in the immunoblotting images.

: We included protein markers in Supplementary Figure 2a. The band size of BACH1 was also indicated.

15. Figure 3c and f, upper and lower panels should be explained in the figure legends.

: We included more details about the invasion assay data in Figure 3c, d, and f in the legend, indicating upper panels are from collagen assay and lower ones are from the invasion chamber assay.

16. On line 96 "OS, $p=0.0194149$; MFS, $p=0.007587$ " are not consistent with the values in the figure.

: We apologize for the inconsistency. We corrected the p values as OS, $p=0.0029$; MFS, $p=0.0190$.

17. On lines 140 and 206, what other cells the expression levels of Nrf2 were compared with?

: We have further clarified how NRF2 expression was compared. As we included in supplementary Figure 2c, 3e, and 3f, NRF2 expression is very low in breast cancer cells compared to OVCAR8 and A549. Low expression of NRF2 in breast cancer cells is already published in the paper by Loignon et al. (Mol. Cancer. Therapy. 2009). Since lung cancer cells including A549 are known to highly express NRF2 due to KEAP1 mutations and hypermethylation (Taguchi et al. 2008 and Wang et al. 2008), we compared our breast cancer cell results with A549. We also found that OVCAR8 expresses high levels of NRF2 when compared with breast cancer cells. Therefore, we used these two cells to compare the role of MAFF in the presence of NRF2. NRF2 expression in other cell lines was also determined and the data are added to Supplementary Figure 2f.

18. On line 171, how long was the hypoxia treatment? 16 hrs in the text but 24 hrs in the figures.

: We corrected it to "24 hours" in the manuscript on page 8 since we treated cells under hypoxia for 24 hours to be consistent with our initial finding of MAFF using microarray (Krieg et al. 2010. Mol Cell Biol).

19. On line 244, cite figure 7b.

: We now cite "Figure 7b" on page 11.

20. On lines 250-255, figure panels are wrongly cited.

: We have correctly cited figure numbers with newly added figures of the rescue experiment by overexpressing IL11.

21. On line 315, mediator not mediator.

: We corrected it to “mediator” in the manuscript on page 16.

21. On line 459, provide programs for testing. In the figure legends, describe which tests were used.

: We used Graphpad Prism to perform statistical tests. We also added these details in the Materials and Methods section and figure legends.

22. Fig. 1c, GO rather than GEO.

: We corrected it to GEO as suggested.

23. Fig. 2 and other figures, use unified names for genes or proteins.

: We have corrected names of genes and proteins.

24. Fig. S2c, values of expression amounts in these cells are lost by dividing hypoxia values with normoxia values.

: As you suggested, to show differences in MAFF basal mRNA levels in each cell line, we included data which were not normalized to normoxia values. Now modified data are included in Figure 2a.

26. Fig. 3e,e,f, brief description of culture conditions in the legend will help understanding.

: We included more details in the Figure 3 legend and the Methods section.

27. Fig. 4d, how were tumor weights determined? Also, representative pictures will help.

: As our group previously published (Kariolis et al. 2014 and 2016), 28 days after ovarian cancer cells were intraperitoneally injected, we sacrificed mice to determine metastatic burden. All visible metastasis in the peritoneal cavity were excised and weighed after counting. These details are now included in the Materials and Methods section. We also included a representative image of our *in vivo* ovarian cancer model in Figure 4d.

28. Fig. 5, ChIP-QPCR to examine increased binding of MAFF to some of the target genes should be shown.

: We included ChIP-qPCR results of MAFF and BACH1 binding to the IL11 promoter, which was identified by ChIP seq in Figure 7e.

29. No legend for Fig. 6e.

: We now include the legend for modified Figure 6.

30. Fig. 7f, Do IL11 high and low groups show any difference? This is important to establish the suggested role of IL11 in tumor metastasis and prognosis of patients.

: From the dataset that we used, we could not detect prognostic significance of IL11 in predicting patient survival. Since IL11 can be induced by other factors, this is somehow expected. However, when we combined MAFF and IL11 expression, their expression was a strong prognostic factor for poor patient survival, indicating that MAFF specific regulation of IL11 is an important driving factor for tumor aggressiveness.

31. Fig. s1, no description of statistical test in the legend.

: We now include the description of the statistical test in the legend,

32. Fig. s6b, did MAFF knockdown remain effective after IL11 transfection? RT-PCR?

: As included in Figure 6e, IL11 overexpression did not change MAFF expression in cells with or without MAFF knockdown.

Kazuhiko Igarashi

Reviewer #3 (Remarks to the Author):

The authors of the study entitled "The HIF target MAFF promotes tumor invasion and metastasis through IL11 and STAT3 signaling" start their work from a RCC4 clear-cell renal cell carcinoma line screen on hypoxia-relevant genes potentially changing metastasis. They evaluate publically available data to investigate the most significant expression changes of candidate genes in various solid cancers. Subsequently, due to negative prognostic values they focus on the transcription factor MAFF in breast cancer (BCa) that lacks a transactivation domain, but functions in association with coactivator or corepressor, where they identify in a nice MS approach following affinity chromatography interaction with BACH. The authors conclude that a BACH1-MAFF interaction is crucial for invasive properties of BCa cell lines and this cell line in vitro work insight is solid. **Less solid is the lack of proliferation change and a too limited in vitro cell line analysis where the reviewer suggest more insightful doable experimentation.** Overall, the study has mechanistic power and it is relevant with illuminating the clinical dilemma of metastasis. **A drawback comes from experimental systems that rely heavy on cancer cell lines and their genetic manipulation, sometimes experimental setup to reveal proof for metastasis is not really state-of-the-art, questioning relevance.** Here, more should be done for a Nature Communication article to substitute the major conclusion that the BACH1-MAFF interaction is

crucial for metastasis. **To extend on in vitro analysis a simple analysis on lactate and pyruvate levels in cancer cell lines should be undertaken and the metabolic controls of these proteins could be better placed in context for writing the discussion. Last but not least, IL-11 autocrine loops might be a too narrow view and IL-6 trans-signaling could very well be relevant.** In summary, the study could benefit from doable experiments, plus an improved discussion before publication.

Major:

- 1) Cell line work allows for cancer cell metabolism insights and that could influence e.g. through mTOR signalling and S6 kinase steering cell seize control. STAT3/5 transcription factors can induce expression of cytokines, growth factors, chemokines or their cognate receptor chains, thereby regulating innate and acquired immunity or to promote tumor surveillance. The authors focus here on hypoxia gene expression also changing tumor vasculature and in this regard **it is important to discuss that HIF1 α / β /VEGF family members can be under control of STAT3/5 transcription factors in breast cancer cells. Furthermore, metabolic control by serine phosphorylated STAT3 downstream of RAS transformation or oncogenic RAS signaling in general will be switched on in most if not all of these used carcinoma cell lines.** This will probably promote through mitochondrial function of serine phosphorylated pSSTAT3 or through transcriptional activation of tyrosine phosphorylated pYSTAT3 it (especially if mitochondria are damaged and not functional). One would expect **significant changes in levels of pyruvate and lactate production leading to promotion of angiogenesis dependent on normoxic or hypoxic conditions**, associated with the Warburg effect, change of lactate transporter genes, etc. promoting at the end the surviving of cancer cells under hypoxic conditions and promoting tumor neovasculature. The authors should at least measure upon **genetic manipulation with high or low MAFF expression levels of pyruvate and lactate production in their tissue culture systems.** Normally, proliferation is associated with high metabolism rates and arrest of proliferation was reported to happen for metastasis, the authors report no change in growth of their cancer cell line systems, but the reviewer is not so convinced here that a metabolic reprogramming does not happen here. Metastatic cells could lose contact to the plastic plate, subsequently go into solution and die quickly off, authors might miss it which could lead to misinterpretation of results. **Here, paired model BCa cell lines under normoxic or hypoxic conditions are suitable to get an insight on simple lactate and pyruvate levels that are under control of STAT3 and should be changed to the understanding of the reviewer under hypoxic or normoxic conditions. Involvement of metabolic change by seahorse measurement is**

also a possible alternative readout. Possible change of metabolism could be better placed in experimental context as well as discussed.

: As you suggested, to determine whether MAFF plays a role in metabolism despite the lack of its influence on tumor cell growth, we have measured lactate and pyruvate levels under normoxia and hypoxia with or without MAFF expression. As included in Supplementary Figure 3c, we have observed that MAFF knockdown did not significantly change lactate production either under normoxia or hypoxia, although hypoxia increased lactate levels, which is well known. This observation was consistent with our Seahorse Analyzer experiments measuring extracellular acidification rates (ECAR) in Supplementary Figure 3d. Interestingly, we found that knocking down MAFF decreased pyruvate in cell media, which is consistent with a previous publication by Lee et al. (2019. Nature) showing that BACH1 regulates mitochondrial metabolism. However, it is beyond the scope of this study to evaluate the role of MAFF in mitochondrial metabolism, which will be a topic for future studies.

2) **The authors perform not really a metastasis assay that is physiologic and relevant.** A changed larger cell size would influence (particularly in an artificial setup like here with tail vein injection) invasion or clogging of BCa cells in lung vessels. In contrast smaller cell size would lead to better clearance. A better metastasis read out for the crucial MAFF genetic cell line pair could be investigated, changed mechanical properties of injected controlled cell size cell line pairs need to be shown. At the end blood vessels in the lung are the smallest and e.g. if cancer cells have different size (in association with suspected change in metabolism), which could e.g. be controlled by simple forward side scatter analysis in a FACS machine that used cell lines are indeed similar of size profiles then the largest carcinoma cells get blocked in the smallest vessels by mechanical principles. That is not really a physiologic BCa metastasis assay, the authors should look up e.g. a recent Nature paper of STAT3 role in liver metastasis for a GEMM(<https://doi.org/10.1038/s41586-019-1004-y>) in systems of carcinogenesis how animal model set up can be investigated.

In BCa research a quick read out animal model system are e.g. Rag2^{-/-}γc^{-/-} or NSG mice on a Balb/C background, which are susceptible for BCa metastasis assays in short read out times. One could e.g. take human BCa cell lines such as e.g. 1x10⁶ MDA-MB-231 could be injected into the 4th mammary fat pad of female immunosuppressed mice. Here authors should make use of their paired MAFF genetic cell system with stable high or low MAFF expression. Alternatively, also an **immunocompetent animal model** that relies on transplant can be used e.g. with 1x10⁴ 4T1 cells (where authors should genetically manipulate MAFF expression) injected into the 4th mammary fat pad of female wildtype Balb/C mice. When tumors reach about 100 mm³ in volume, the

primary tumors need to be resected. Ten days after primary tumor removal, mice need to be sacrificed, lungs need to be isolated and the lung weight should be assessed as well as **lung pathology using specific epithelial and EMT cell markers would be most insightful, where e.g. VIMENTIN, CK8, KI-67, CLEAVED CASPASE-3 Hematoxylin and Eosin G should be carried out** for minimal histo-pathology analysis to provide a more physiologic BCa test system.

: To determine the role of MAFF in a more physiologically relevant *in vivo* model, we injected MDA-MB-231 cells into the mammary fat pad of NSG mice, which are known to form spontaneous lung metastasis within 2 months¹. Consistent with our observations using a tail vein injection model, we found that the MAFF knockdown group showed less lung metastasis. We also included an IL11 overexpression group with MAFF knockdown and found that IL11 was able to rescue the decreases in lung metastasis by MAFF knockdown, suggesting that IL11 plays a role as a downstream target of MAFF. Interestingly, we observed that lung nodules from the spontaneous model showed features of micrometastasis rather than bigger nodules observed from the tail vein injection model, and these features made them hard to stain for EMT cell markers in lung. Although a recent study showed that BACH1 is involved in EMT to regulate tumor metastasis³, we did not have any indication that MAFF was a critical regulator of EMT in the studies that we performed. Also, the role of MAFF in EMT was not the focus of this manuscript. Instead we focused on how MAFF regulates tumor angiogenesis through IL11-STAT3 pathways. However, MAFF and EMT will be the subject of an independent study in the future.

Minor:

- 2) A model could be drawn in the last part of the study where the reviewers suggest to draw an oncogenic transcription factor network such as MAFF-BACH interaction, STAT3 and HIF1 α / β could be interesting to readers to present in a scheme, e.g. summarizing findings in a cartoon at the last Figure part.

: We made a cartoon to summarize our findings and included it in Figure 9.

4) With regard to the interpretation of the xenograft experiments presented in this study it is important to note that IL-6 trans-signaling as well as IL-11 activity are both active in mice or patient derived cells giving rise to xenograft tumors in immunosuppressed mice allowing for corresponding cytokine-receptor interaction [Jostock, T., Mullberg, J., Ozbek, S., Atreya, R., Blinn, G., Voltz, N., et al., (2001) Soluble gp130 is the natural inhibitor of soluble interleukin-6 receptor transsignaling responses. Eur J Biochem. 268(1): p. 160-7.]. Therefore, most likely activators of STAT3 signaling in mouse xenografts are not restricted to IL-11 only and will originate also from IL-6 trans-signaling. **The authors should therefore at least discuss plasma soluble IL-6R α**

levels and take that into account. All other gp130 cytokine ligands are species restricted. The authors have autocrine cytokine action evidence of IL-11 secretion and signaling in BCa cells, but IL-6 trans-signaling could enhance the vicious cycle, subsequently driving HIF-1alpha and MAFF expression.

: IL6 is also an important factor regulating the STAT3 pathway and can also be induced under hypoxia. From our RNA-seq and ChIP-seq data, we did not find IL-6 was significantly altered by MAFF knockdown. However, to confirm whether there is any existing feedback loop regulating the IL-6/STAT3 pathway, we measured IL-6 levels in mouse serum levels using an ELISA assay. Compared to IL11, which showed consistent results in our studies, we did not observe any significant changes in IL6 levels. Data are included in Figure 8b.

5) The sentence on page 7 "Overlapping regions of MAFF and pimonidazole,...." is misleading and could be reworded/improved for understanding as follows: "Overlapping regions of MAFF and hypoxic areas as labelled by the chemical pimonidazole as a hypoxic tumor area marker were frequently observed as quantified by histology scores (~87%)...."

: The sentence has been modified to read, "Overlapping regions of MAFF and hypoxic areas, which were identified using the hypoxia marker pimonidazole, were frequently observed (~87% colocalization)," supporting a link between hypoxia and MAFF expression in tumors (Supplementary Fig. 4a).

Reviewer #4 (Remarks to the Author):

Comment:

Hypoxia plays a critical role in tumor progression including invasion and metastasis. The manuscript titled "The HIF target MAFF promotes tumor invasion and metastasis through IL11 and STAT3 signaling" by Eui Jung Moon et al. screened fifty hypoxia inducible genes, identified MAFF as a potent regulator of tumor invasion without affecting cell viability, which is elevated in metastatic breast cancer patients and is specifically correlated with hypoxic tumors. The authors further identified IL11 as a direct transcriptional target of the heterodimer between MAFF and BACH1, leading to the activation of STAT3 signaling, demonstrating the oncogenic role of MAFF as an activator of the IL11/STAT3 pathways in breast cancer. The manuscript is very well written and is well balanced. The story is interesting, and the findings are significant. However, several concerns need to be answered before considering acceptance.

Major points:

1. It lacks in vivo data to support the importance of IL11-STAT3 pathway in MAFF induced tumor cell invasion and metastasis.

: We repeated our in vivo experiments by injecting MDA-MB-231 cells into the mammary fat pad of NSG mice to form spontaneous lung metastasis. In this newly added study, we also included a group of MAFF knockdown with IL11 overexpression to demonstrate a role of IL11 in tumor metastasis as a downstream target of MAFF. Activation of the STAT3 pathway was also determined by phospho-specific STAT3 staining of tumor tissues. Data are included in Figure 8.

2. The effect of MAFF in blood vessel development should be measured in tumor tissues. The result of knockout of IL11 is not strong enough to support MAFF's role in blood vessel development.

: To further address the role of IL11 in angiogenesis *in vivo* settings, we stained tumor tissues for MECA32, an endothelial cell marker. As we observed from the study, MAFF knockdown decreased MECA32 staining. In addition, overexpression of IL11 increased MECA32 staining in MAFF knockdown tumors. This data is included in Figure 8c.

3. Since inhibition of MAFF suppresses tumor invasion without affecting cell viability, and the author concluded that the combined regulation of IL11-STAT3 signaling pathway by MAFF and BACH1 is responsible for MAFF induced tumor cell invasion and metastasis. However, there is not enough proof in this manuscript to show the inhibition of BACH1, or IL11-STAT3 could suppress tumor invasion without affecting cell viability.

: To determine whether BACH1, IL11, and STAT3 are involved in tumor invasion without affecting cell viability, we measured tumor cell growth after knocking down these genes. As we found with MAFF knockdown, inhibition of these genes did not change tumor cell growth, indicating they do not play a role in cell viability in our experimental settings. Data are now included in Supplementary Figure 8a.

Minor points:

1. Please describe how the survival analysis was performed in Fig 1b in detail.

: We added more details on how we determined patient survival in Figure 1b. Expression of these 50 hypoxic-inducible genes were combined and analyzed using the prognostic database, ProgGeneV2². We have included the sentence, "Using prognostic database, ProgGeneV2², we found that the combined expression of these genes correlated with poor patient survival, suggesting an oncogenic role for them in breast cancer (GSE19783) (Fig. 1b)" on page 4.

2. Please explain how the intensity was measured in Fig 1e IHC data. Expression data of MAFF in primary tumor and lymph node from online dataset may be added.

: We have included additional details in the Methods section on how we measured MAFF staining intensity. "MAFF staining in TMAs was evaluated using FIJI (Image-J-based open-source software)⁴. The color deconvolution feature was used to extract the positively stained areas. Integrated density values were calculated from the resulting binary image."

3. The analysis in Fig 1f lacks p value and R value.

: We apologize for the omission. Now we added p and R values for Figure 1f.

4. In Fig 4b, please replace "MVD" with "MECA-32".

: We changed MVD to MECA-32 in Figure 4b.

5. For Fig 5c, it is not clear the analysis was performed with 240 genes significantly altered under hypoxia or 160 genes regulated in both normoxia and hypoxia. Please clearly clarify it.

: We performed DAVID analysis using 415 genes regulated under normoxia and 240 genes under hypoxia. We changed the sentence in the manuscript to read, "To obtain a global overview of the biological functions of proteins encoded by genes identified from RNA seq either under normoxia or hypoxia or both..." on page 8.

6. The table in Fig 7c lacks p value.

: We have included a p value in Figure 7c.

7. Correlation analysis of HIF-MAFF, MAFF, BACH1/IL11 is needed with expression data in tumor tissues.

: Using primary tumor tissues from our *in vivo* studies, we stained MAFF, HIF-1, BACH1, and IL11. Their correlation was determined by evaluating colocalization of staining from tissues using ImageJ. Data are included in Supplementary Figure 9d.

1 Iorns, E. *et al.* A new mouse model for the study of human breast cancer metastasis. *PLoS One* **7**, e47995, doi:10.1371/journal.pone.0047995 (2012).

2 Goswami, C. P. & Nakshatri, H. PROGeneV2: enhancements on the existing database. *BMC Cancer* **14**, 970, doi:10.1186/1471-2407-14-970 (2014).

- 3 Sato, M. *et al.* BACH1 promotes pancreatic cancer metastasis by repressing epithelial genes and enhancing epithelial-mesenchymal transition. *Cancer Res*, doi:10.1158/0008-5472.CAN-18-4099 (2020).
- 4 Schindelin, J. *et al.* Fiji: an open-source platform for biological-image analysis. *Nat Methods* **9**, 676-682, doi:10.1038/nmeth.2019 (2012).

Reviewers' comments:

Reviewer #1 (Remarks to the Author):

The authors have satisfactorily responded to my concerns. I have no further hesitation about publication.

Reviewer #2 (Remarks to the Author):

The authors nicely addressed most of the comments by this reviewer. However, one issue remains unclear. Authors now show ChIP-seq data (distribution of tags) in Fig. 5f, which is not persuasive enough. The presumptive peaks of MAFF are almost the same as the background distributions. Authors need to show tag distribution of input samples and how many peaks were identified by the statistical analysis by the program they used. Are these numbers within expected range for transcription factors? Also, it becomes now important to show peak patterns of known target genes of MAFF like Hmox1. The numbers of genes in the text (lines 205-206) are not consistent with the figure 5e and the legend.

Kazuhiko Igarashi

Reviewer #3 (Remarks to the Author):

The authors improved the experimental validation system to the breast cancer metastasis assays questioned by several reviewers using mammary fat pad injection into NSG mice with controlled cell line systems with or without genetic manipulation of key components as illuminated in the study. This has improved the power of the study and it recapitulated major findings strengthening overall conclusions. Moreover, a number of other reviewer concerns or controls were made, missing information or corrections were done.

The question of IL-6 trans-signaling is not answered by measuring IL-6 and the conclusion is here wrong, since one has to measure IL-6Ralpha chain cleavage by ADAM proteases. Carcinomas secrete or express in their extracellular matrix proteases that cleave from incoming myeloid cells associated with inflammatory processes the IL-6Ralpha chain, that then binds some IL-6 and is 10 more potent

compared to IL-6 alone. Very relevant in rodent models, particular relevant in NSG mice that have intact myeloid system and tumor functions. This might go beyond this manuscript, but the conclusion should be corrected and the authors are advised to read up on literature in breast cancer or other carcinoma types what the role and function of IL-6 trans-signaling is and how the system works, to be more flexible on concluding better.

Reviewer #4 (Remarks to the Author):

This revised version answered all previous questions/comments very well and demonstrated that MAFF's important role in breast cancer metastasis with new xenograft assay.

I do not have any additional questions.

We have addressed Reviewers' comments below.

Reviewer #2 (Remarks to the Author):

The authors nicely addressed most of the comments by this reviewer. However, one issue remains unclear. Authors now show ChIP-seq data (distribution of tags) in Fig. 5f, which is not persuasive enough. The presumptive peaks of MAFF are almost the same as the background distributions. Authors need to show tag distribution of input samples and how many peaks were identified by the statistical analysis by the program they used. Are these numbers within expected range for transcription factors? Also, it becomes now important to show peak patterns of known target genes of MAFF like *Hmox1*. The numbers of genes in the text (lines 205-206) are not consistent with the figure 5e and the legend.

Kazuhiko Igarashi

: We want to thank the reviewer for this follow-up on our ChIP-seq data includes a high background signal. The reasons for this background are unclear. For this reason, we made extra effort to confirm MAFF regulation of *IL11* through qPCR, ChIP, and luciferase reporter assays. We also carefully searched for sequences of the MAF response element (MARE) and antioxidant response element (ARE) in MAFF binding regions. When compared to *GCLM*, a known small MAF target gene^{1,2}, we found MARE or ARE sequences near the identified regions of MAFF binding in all identified genes, indicating these genes are regulated by MAFF based on ChIP-seq and RNA-seq results. Now data including peaks of MAFF binding and input are included in Fig. 5f and Supplementary Fig. 6 with highlighted MARE or ARE sequences. We also included the following sentences on page 9-10 of the main text for clarity.

“We further confirmed that the MAFF binding peaks in the *IL11* promoter included the MARE/ARE sequence (TGAGCTCGC) using the UCSC genome browser (Figure 5f). When compared with *GCLM*, a known target of small MAF proteins^{1,2}, we identified other genes that also possess MARE or ARE sequences in their MAFF binding regions (Supplementary Figure 6).”

Reviewer #3 (Remarks to the Author):

The authors improved the experimental validation system to the breast cancer metastasis assays questioned by several reviewers using mammary fat pad injection into NSG mice with controlled cell line systems with or without genetic manipulation of key components as illuminated in the study. This has improved the power of the study and it recapitulated major findings strengthening overall conclusions. Moreover, a number of other reviewer concerns or controls were made, missing information or corrections were done.

The question of IL-6 trans-signaling is not answered by measuring IL-6 and the conclusion is here wrong, since one has to measure IL-6Ralpha chain cleavage by ADAM proteases. Carcinomas secrete or express in their extracellular matrix proteases that cleave from incoming myeloid cells associated with inflammatory processes the IL-6Ralpha chain, that then binds some IL-6 and is 10 more potent compared to IL-6 alone. Very relevant in rodent models, particular relevant in NSG mice that have intact myeloid system and tumor functions. This might go beyond this manuscript, but the conclusion should be corrected and the authors

are advised to read up on literature in breast cancer or other carcinoma types what the role and function of IL-6 trans-signaling is and how the system works, to be more flexible on concluding better.

:We want to thank the reviewer for making this point and we reviewed the literature on IL-6 trans-signaling. To address the possible involvement of IL6 pathway through trans-signaling, we have included a paragraph in the Discussion section of the manuscript.

“In our *in vivo* studies, we also showed that MAFF knockdown decreased tumor lung metastasis in both tail vein injection and spontaneous metastasis models (Fig. 4 and 8). In the orthotopic mouse model, we also confirmed that overexpression of IL11 rescued the decreased tumor metastasis and expression of phospho-specific STAT3 expression in the absence of MAFF, supporting a role of IL11 in MAFF-mediated tumor metastasis (Figure 9). However, although we did not observe changes in human IL6 from mouse serum carrying MDA-MB-231 with or without MAFF expression, we still need to consider the effect of IL6 pathways through trans-signaling. While the classic IL6 pathway is activated through the binding of IL6 to IL6 receptor (IL6R) on the cell surface, proteolytic cleavage of IL6R by ADAM17 or ADAM10 or alternative splicing of IL6R mRNA results in trans-signaling of IL6 through soluble IL6R (sIL6R)³. Then interaction of sIL6R and gp130 activates STAT3 pathways in cells that do not express IL6R. Since ADAM10 and ADAM17 are hypoxia targets, we cannot rule out the possible involvement of sIL6R in elevated STAT3 signaling^{4,5}. Also, crosstalk between human and murine IL6 pathways in the murine tumor microenvironment by tumor infiltrating immune cells or fibroblasts can potentially prime the metastatic niche and hyperactivate STAT3 pathways in cancer cells through IL6 trans-signaling pathways^{6,7}. Therefore, further studies are required to more rigorously determine the role of IL6 in HIF-1-MAFF-STAT3 pathways.”

- 1 Warnatz, H. J. *et al.* The BTB and CNC homology 1 (BACH1) target genes are involved in the oxidative stress response and in control of the cell cycle. *J Biol Chem* **286**, 23521-23532, doi:10.1074/jbc.M111.220178 (2011).
- 2 Hirotsu, Y. *et al.* Nrf2-MafG heterodimers contribute globally to antioxidant and metabolic networks. *Nucleic Acids Res* **40**, 10228-10239, doi:10.1093/nar/gks827 (2012).
- 3 Johnson, D. E., O'Keefe, R. A. & Grandis, J. R. Targeting the IL-6/JAK/STAT3 signalling axis in cancer. *Nat Rev Clin Oncol* **15**, 234-248, doi:10.1038/nrclinonc.2018.8 (2018).
- 4 Barsoum, I. B. *et al.* Hypoxia induces escape from innate immunity in cancer cells via increased expression of ADAM10: role of nitric oxide. *Cancer Res* **71**, 7433-7441, doi:10.1158/0008-5472.CAN-11-2104 (2011).
- 5 Szalad, A., Katakowski, M., Zheng, X., Jiang, F. & Chopp, M. Transcription factor Sp1 induces ADAM17 and contributes to tumor cell invasiveness under hypoxia. *J Exp Clin Cancer Res* **28**, 129, doi:10.1186/1756-9966-28-129 (2009).
- 6 Garbers, C. *et al.* Species specificity of ADAM10 and ADAM17 proteins in interleukin-6 (IL-6) trans-signaling and novel role of ADAM10 in inducible IL-6

receptor shedding. *J Biol Chem* **286**, 14804-14811, doi:10.1074/jbc.M111.229393 (2011).

- 7 Jostock, T. *et al.* Soluble gp130 is the natural inhibitor of soluble interleukin-6 receptor transsignaling responses. *Eur J Biochem* **268**, 160-167, doi:10.1046/j.1432-1327.2001.01867.x (2001).

We have addressed Reviewers' comments below.

Reviewer #2 (Remarks to the Author):

The authors nicely addressed most of the comments by this reviewer. However, one issue remains unclear. Authors now show ChIP-seq data (distribution of tags) in Fig. 5f, which is not persuasive enough. The presumptive peaks of MAFF are almost the same as the background distributions. Authors need to show tag distribution of input samples and how many peaks were identified by the statistical analysis by the program they used. Are these numbers within expected range for transcription factors? Also, it becomes now important to show peak patterns of known target genes of MAFF like Hmox1. The numbers of genes in the text (lines 205-206) are not consistent with the figure 5e and the legend.

Kazuhiko Igarashi

: We want to thank the reviewer for this follow-up on our ChIP-seq data. Please understand that our responses were significantly delayed to the pandemic and stay-in-place order, which limit our access to a laboratory and core facilities to perform experiments. We also encountered some technical difficulties to repeat MAFF ChIP-sequencing. Therefore, we decided to perform V5-ChIP sequencing using V5-tagged MAFF expressing MDA-MB-231 cells after knocking down endogenous MAFF. As described in our Methods section, peak calling was performed using MACS. We identified that 40.44% of MAFF binding peaks were in promoter region while 27.36% were located in intron. The analysis of DNA binding motif using MEME also found significant binding on similar sequences as known MAFF binding motif.

In addition, data mining using DAVID analysis and PubMatrix showed that most of MAFF target genes regulating tumor invasion and metastasis were overlapped with our previous ChIP-seq results. In addition to our initial finding, we also showed that TGFB2 and TNC were bound and regulated by MAFF under both normoxia and hypoxia. However, our qRT-PCR data confirmed that only the expression of IL11 was significantly altered by MAFF and hypoxia.

We also included data showing peaks and distribution tags of both MAFF IP and input samples in Fig. 5h.

Although knocking down MAFF did not significantly change expression of *HMOX1*, a well-known target gene of NRF2 and BACH1 through their competitive binding², we found significant peaks in two *HMOX1* promoter regions. We also validated a strong peak on *IL11* promoter, which includes the same MAFF binding sequence (GCGAGCTCA) as the one that we found from the initial MAFF ChIP-sequencing.

With newly performed ChIP-sequencing data, now we included the following sentences on page 8-9 of the main text for clarity.

“To further identify genes that are directly regulated by MAFF through transactivation, we performed ChIP-sequencing using V5-tagged MAFF overexpressing MDA-MB-231 after knocking down endogenous MAFF. Global analysis of all peaks for MAFF revealed that 40.44% in promoter and 27.36% in intron regions (Fig. 5e). From MEME analysis, a MAFF enriched motif was found that is similar with the consensus MARE sequence (Fig. 5f). By combining profiles from RNA-sequencing and ChIP-sequencing, we highlighted 358 genes under normoxia, 136 genes under hypoxia, and among those 66 genes under both conditions that were directly bound and regulated by MAFF (Fig. 5g). We further identified MAFF regulated genes that were specifically involved in tumor cell invasion and metastasis using DAVID and Pubmatrix, a text mining tool (Supplementary Fig. 5a)^{22,23}. Among these genes, under both normoxia and hypoxia, 3 (*TGFB2*, *BMP4*, *COL1A1*) were upregulated when MAFF was knocked down, and 2 genes (*IL11*, *TNC*) were downregulated when MAFF was inhibited, suggesting MAFF negatively or positively regulates expression of these genes. RNA-sequencing results were validated by qRT-PCR and only *IL11* and *TGFB2* showed similar changes by MAFF knockdown and hypoxia (Supplementary Fig. 5b). In OVCAR8 cells, which highly express NRF2, loss of MAFF resulted in inhibition of *IL11*, while rest of genes were not significantly changed (Supplementary Fig. 5c).

We further searched the MAFF binding peaks in target gene promoters using the UCSC genome browser (Figure 5g). Although gene expression of *HMOX1* was not significantly altered by MAFF knockdown, we found binding peaks on two known promoter regions²⁴. Likewise, we identified a strong peak on promoter region of *IL11* possessing MARE/ARE sequence (GCGAGCTCA), indicating that MAFF directly regulates *IL11* transcription.”

Reviewer #3 (Remarks to the Author):

The authors improved the experimental validation system to the breast cancer metastasis assays questioned by several reviewers using mammary fat pat injection into NSG mice with controlled cell line systems with or without genetic manipulation of key components as illuminated in the study. This has improved the power of the study and it recapitulated major findings strengthening overall conclusions. Moreover, a number of other reviewer concerns or controls were made, missing information or corrections were done.

The question of IL-6 trans-signaling is not answered by measuring IL-6 and the conclusion is here wrong, since one has to measure IL-6Ralpha chain cleavage by ADAM proteases. Carcinomas secrete or express in their extracellular matrix proteases that cleave from incoming myeloid cells associated with inflammatory processes the IL-6Ralpha chain, that then binds some IL-6 and is 10 more potent compared to IL-6 alone. Very relevant in rodent models, particular relevant in NSG mice that have intact myeloid system and tumor functions. This might go beyond this manuscript, but the conclusion should be corrected and the authors are advised to read up on literature in breast cancer or other carcinoma types what the role and function of IL-6 trans-signaling is and how the system works, to be more flexible on concluding better.

:We want to thank the reviewer for making this point and we reviewed the literature on IL-6 trans-signaling. To address the possible involvement of IL6 pathway through trans-signaling, we have included a paragraph in the Discussion section of the manuscript.

“In our *in vivo* studies, we also showed that MAFF knockdown decreased tumor lung metastasis in both tail vein injection and spontaneous metastasis models (Fig. 4 and 8). In the orthotopic mouse model, we also confirmed that overexpression of IL11 rescued the decreased tumor metastasis and expression of phospho-specific STAT3 expression in the absence of MAFF, supporting a role of IL11 in MAFF-mediated tumor metastasis (Figure 9). However, although we did not observe changes in human IL6 from mouse serum carrying MDA-MB-231 with or without MAFF expression, we still need to consider the effect of IL6 pathways through trans-signaling. While the classic IL6 pathway is activated through the binding of IL6 to IL6 receptor (IL6R) on the cell surface, proteolytic cleavage of IL6R by ADAM17 or ADAM10 or alternative splicing of IL6R mRNA results in trans-signaling of IL6 through soluble IL6R (sIL6R)⁵. Then interaction of sIL6R and gp130 activates STAT3 pathways in cells that do not express IL6R. Since ADAM10 and ADAM17 are hypoxia targets, we cannot rule out the possible involvement of sIL6R in elevated STAT3 signaling^{6,7}. Also, crosstalk between human and murine IL6 pathways in the murine tumor microenvironment by tumor infiltrating immune cells or fibroblasts can potentially prime the metastatic niche and hyperactivate STAT3 pathways in cancer cells through IL6 trans-signaling pathways^{8,9}. Therefore, further studies are required to more rigorously determine the role of IL6 in HIF-1-MAFF-STAT3 pathways.”

References

- 1 Landt, S. G. *et al.* ChIP-seq guidelines and practices of the ENCODE and modENCODE consortia. *Genome Res* **22**, 1813-1831, doi:10.1101/gr.136184.111 (2012).
- 2 Reichard, J. F., Motz, G. T. & Puga, A. Heme oxygenase-1 induction by NRF2 requires inactivation of the transcriptional repressor BACH1. *Nucleic Acids Res* **35**, 7074-7086, doi:10.1093/nar/gkm638 (2007).
- 3 Hirotsu, Y. *et al.* Nrf2-MafG heterodimers contribute globally to antioxidant and metabolic networks. *Nucleic Acids Res* **40**, 10228-10239, doi:10.1093/nar/gks827 (2012).
- 4 Warnatz, H. J. *et al.* The BTB and CNC homology 1 (BACH1) target genes are involved in the oxidative stress response and in control of the cell cycle. *J Biol Chem* **286**, 23521-23532, doi:10.1074/jbc.M111.220178 (2011).
- 5 Johnson, D. E., O'Keefe, R. A. & Grandis, J. R. Targeting the IL-6/JAK/STAT3 signalling axis in cancer. *Nat Rev Clin Oncol* **15**, 234-248, doi:10.1038/nrclinonc.2018.8 (2018).
- 6 Barsoum, I. B. *et al.* Hypoxia induces escape from innate immunity in cancer cells via increased expression of ADAM10: role of nitric oxide. *Cancer Res* **71**, 7433-7441, doi:10.1158/0008-5472.CAN-11-2104 (2011).
- 7 Szalad, A., Katakowski, M., Zheng, X., Jiang, F. & Chopp, M. Transcription factor Sp1 induces ADAM17 and contributes to tumor cell invasiveness under hypoxia. *J Exp Clin Cancer Res* **28**, 129, doi:10.1186/1756-9966-28-129 (2009).
- 8 Garbers, C. *et al.* Species specificity of ADAM10 and ADAM17 proteins in interleukin-6 (IL-6) trans-signaling and novel role of ADAM10 in inducible IL-6 receptor shedding. *J Biol Chem* **286**, 14804-14811, doi:10.1074/jbc.M111.229393 (2011).
- 9 Jostock, T. *et al.* Soluble gp130 is the natural inhibitor of soluble interleukin-6 receptor transsignaling responses. *Eur J Biochem* **268**, 160-167, doi:10.1046/j.1432-1327.2001.01867.x (2001).

REVIEWERS' COMMENTS

Reviewer #2 (Remarks to the Author):

Authors addressed effectively the issue this reviewer raised before. Congratulations on a very important, breakthrough paper. When finalizing the paper, please consider providing the genes in Fig. 5g as a supplementary table, which will be very beneficial to other investigators.

Kazuhiko Igarashi

We have addressed Reviewers' comments below.

Reviewer #2 (Remarks to the Author):

Authors addressed effectively the issue this reviewer raised before. Congratulations on a very important, breakthrough paper. When finalizing the paper, please consider providing the genes in Fig. 5g as a supplementary table, which will be very beneficial to other investigators.

Kazuhiko Igarashi

: Thank you so much for your kind comment. Now we have added the list of genes in Figure 5g as a supplementary table (Supplementary Table 2).